# Tidal flood area mapping in the face of climate change scenarios: case study in a tropical estuary in the Brazilian semi-arid region

Paulo Victor N. Araújo[1,2,3], Venerando E. Amaro[1,3], Leonlene S. Aguiar[3], Caio C. Lima[3,4], and Alexandre B. Lopes[5]

[1]Postgraduate Program in Geodynamics and Geophysics (PPGG), Federal University of Rio Grande do Norte, P.O. Box 1524, Natal-RN, 59078-970, Brazil
[2]Research Group on Environmental Analysis, Modelling and Geoinformation (PAMGEIA), Federal Institute of Education, Science and Technology of Rio Grande do Norte, Macau-RN, 59500-000, Brazil
[3]Laboratory of Geotechnologies, Coastal and Ocean Modelling (GNOMO), Department of Civil Engineering, Federal University of Rio Grande do Norte, Natal-RN, 59078-970, Brazil
[4]Federal Institute of Education, Science and Technology of Rio Grande do Norte, Natal-RN, 59015-000, Brazil
[5]Center of Sea Studies, Federal University of Parana, P.O. Box 61, Curitiba-PR, 83225-976, Brazil

*Correspondence to*: Paulo Victor N. Araújo (paulo.araujo@ifrn.edu.br)

**Abstract.** Previous studies on tidal flood mapping are mostly through continental and/or global scale approaches. Moreover, the few works on local scale perception are concentrated in Europe, Asia, and North America. Here, we present a case study approaching a tidal flood risk mapping application in the face of climate change scenarios in a region with a strong environmental and social appeal. The study site is an estuarine cut in the Brazilian semi-arid region, covering part of two state conservation units, which has been suffering severe consequences from tidal flooding in recent years. In this case study, we used high geodetic precision data (LiDAR DEM), together with robust tidal return period statistics and data from current sea level rise scenarios. We found that approximately 327.60 km² of the estuary is under tidal flood risk and in need of mitigation measures. This case study can serve as a basis for future management actions as well as a model for applying risk mapping in other coastal areas.

## 1 Introduction

Climate change has been associated with various environmental and socioeconomic damages worldwide, with global mean sea level rise (SLR) being one of the main associated phenomena (Nicholls and Cazenave, 2010; IPCC, 2014; Busman et al., 2016; Dangendorf et al., 2019; Bamber et al., 2019). The SLR global mean is occurring at an accelerating rate. Special Report on the Ocean and Cryosphere in a Changing Climate (SROCC) finds that global mean sea levels will most likely rise between 0.95 feet (0.29m) and 3.61 feet (1.1m) by the end of this century., threatening coastal communities and ecosystems worldwide (Nerem et al., 2018; Bamber et al., 2019, IPCC, 2019). SLR will radically redefine the coastline of the 21st century (Taherkhani et al., 2020).

The changes produced by rising and falling the mean sea levels have important implications for the dynamics and morphology of coastal environments, and it is in these environments that a considerable part of the world's population lives (Neumann et al., 2015). Furthermore, it has been causing flooding of natural habitats and coastal infrastructures and consequently, causing

environmental and socioeconomic impacts of varying magnitude (Dwarakisha et al., 2009; IPCC, 2014; Murray et al., 2019). Decades ago, the flooding that used to happen only during a powerful or localized storm now can happen when a steady breeze or a change in coastal current overlaps with high tide, as it occurs, for example in USA (NOAA, 2019).

In past decades, high tide flooding had little impact on coastal communities because our shorelines were not as heavily developed, and sea level was not as high. Today, however, the reach and effect of the tides is changing, and many coastal

towns and cities are already grappling with how best to protect their communities and infrastructure (Dahl et al., 2017).

In Brazil, the current panorama of coastal flooding is worrying. The Brazilian Panel on Climate Change (PBMC) systematized data and information indicating that different regions in Brazil are already experiencing changes in their characteristic climates (PBMC, 2014). These changes are expected to affect the country's natural, human, infrastructure and production systems in a non-uniform manner (Brasil, 2016). The country stands out as the seventh largest nation in the world by coastal population

and the seventh prevalent proportion of coastal zones in low-lying areas (Mcgranahan et al., 2007). About 25% of the Brazilian population lives in the coastal zone and has lately been suffering from the damage caused by the relative SLR (SMC-Brasil, 2018). Approximately 60% of the natural events that hit Brazil from 1948 to 2006 with harmful consequences to the population were related to flooding and/or sea advances (Brasil, 2016). These data, combined with scenarios of rising sea level trends (Easterling et al., 2000; Taherkhani et al., 2020), warn us of the need for (local scale) projections for the next decades to support

the preparation and planning to respond to increasing threat related to SLR. According to Taherkhani et al. (2020), the 21st century will see significant changes to coastal flooding regimes (where present-day extreme-but-rare events become common), which poses a major risk to the safety and sustainability of coastal communities worldwide. Climate risk adaptations involving large infrastructure investments represent difficult decisions and require an accurate information base (Hall et al., 2019; Kulp and Strauss, 2019).

Coastal flooding is becoming more frequent and expensive SLR (Herdman et al., 2018). Dahl et al. (2017) concluded that SLR drives increased tidal flooding frequency at tide gauges along the U.S. East and Gulf Coasts in projections for next years. The Fifth Assessment Report of the United Nations Intergovernmental Panel on Climate Change (IPCC) presented climate scenarios called Representative Concentration Pathway (RCP) (IPCC, 2014). Each RCP was defined on a global scale and considers the historical evolution of several factors, such as the cumulative measure of greenhouse gases emission from all

sources of emission and their total radiative forcing pathway and 2100. Under RCP, the IPCC AR5 projected Global Mean Sea Level (GMSL) by summing the input from physical processes to afford a likely (central 66% probability) estimation of GMSL rise of 0.52-0.98m in the case of unmitigated increase of emissions (RCP 8.5) by 2100, relative to 1986-2005 (Church et al., 2013). Although in all scenarios the projected increase in MSL is a maximum of 1m for the year 2100, they have been designed on global and continental scale. Many studies have analysed the risk arising from these flood scenarios across the

low-lying coastal zone (altitude up to 10m) (Nicholls et al., 2007; Dwarakish et al., 2009; Nicholls and Cazenave, 2010;

Nicholls et al., 2011; Boori et al., 2012; Busman et al., 2016). However, the lack of work on the theme in focus in the South American region, especially in Brazil, is striking. Therefore, the objective of this research is to develop and apply a tidal flood risk mapping, given the current scenarios of rising sea level trends, adopting as a case study the Piranhas-Açu Estuary, the northern portion of the state of Rio Grande do Norte, Brazil.

Finally, this study has an innovative character since it applies a robust data set in an integrated spatial analysis.

## 2 Study area

The tropical watershed of the Pianco-Piranhas-Açu River is located in northeastern Brazil and is the widest watershed of the Northeast Atlantic Eastern Hydrographic Region, with a total area of 43,683 km². Its territory is divided between the states of Paraiba (60%) and Rio Grande do Norte (40%). Fully inserted in a very hot and semi-arid climate territory, the basin presents
concentrated rainfall in a few months of the year (rainy season in late autumn and early winter) and a pattern of strong interannual variability, characterized by alternation between years of above average, regular and consecutive years of below-average values resulting in prolonged droughts and low water availability.

The study area covers the entire Pianco-Piranhas-Açu estuary, represented by a rectangle in the measurements 62 x 35.5 km, corresponding to an area of approximately 2,201 km². This area is an estuarine section comprising parts of 4 municipalities of
the northern coast of the state of Rio Grande do Norte (Porto do Mangue, Carnaubais, Pendencias, and Macau) (Figure 1).

In general, the study area is represented by a Tropical of Equatorial Zone climate, and framed in the semi-arid climate subdomain (Diniz and Pereira, 2015). The region in focus stands out as the driest stretch of the entire Brazilian coast, with an average rainfall of 537.6 mm/year in the city of Macau. The daily temperatures vary from 26°C to 30°C (with a mean temperature of 26.8°C), and the average relative air humidity is 70% (IDEMA, 1999; Diniz and Pereira, 2015; Barbosa et al.,
2018a). In geomorphological terms, there is a wide fluvial-marine plain that constitutes the Coastal Strip (Barbosa et al., 2018b, Costa et al., 2020). The number and extent of various channels present along adjacent large river plains reveal the great influence of oceanic waters on this stretch of the continent, with tidal action being one of the major natural forces responsible for hydrographic control.

The study area is inserted in sheltered coastal region dominate by tide-modified and tide-dominated beaches. (Vital et al.,
2016). In this region, the local tide is semidiurnal, with two high and two low tides, where the average level set as a reference is 1.39 m above the reduction level (RL), as established by the Hydrography and Navigation Directorate (DHN) of the Brazilian Navy (MB). The DHN of MB, the body responsible for maritime monitoring, adopts the so-called reduction level (RL) as a reference for maritime quota. The RL is level that corresponds to the average of low syzygy tides to eliminate the variations of tides and to assure navigators that they will not find any depth less than those represented in the nautical chart. The RL is a
chart datum estimation to the mean lower low water. In the region, there are mean semidiurnal high tides of 2.34 m above the RL (Matos et al., 2019; DHN, 2018). As for marine currents, the region is under the influence of the Southern Equatorial Current that acts throughout the northern coast of Brazil (Diniz et al, 2017).

In general terms, this coastal region has a mosaic of ecosystems marked by mangrove forests, together with extensive ebb tidal delta, exposed and sheltered sandy ocean beaches, barrier island systems, active dune fields, spits and short-term high-sedimentary dynamic tidal channels (Grigio et al., 2006; Amaro et al., 2012; Santos and Amaro, 2013; Busman, et al., 2016; Vital et al., 2016). The industrial sector of the area under study comprises essentially mineral exploration, especially salt, oil, and gas. The extraction of oil and natural gas is a very important activity in the basin and economy of the state of Rio Grande do Norte due to the royalties generated (IDEMA, 2005). Aquaculture and artisanal fishing also play a part in the local economy, as well as irrigated agriculture, shrimp farming, salt marshes and, more recently, typical wind industry infrastructure.

There are two noteworthy state conservation units in the study area: Ponta do Tubarao State Sustainable Development Reserve (RDSEPT) and the Rosado Dunes Environmental Protection Area (APADR). The RDSEPT was created through State Law No. 8,349 of July 18, 2003, and its objectives are to safeguard the traditional way of life and ensure activities based on sustainable exploitation of natural resources, traditionally developed over generations and adapted. local ecological conditions, which play a fundamental role in protecting nature and maintaining biological diversity. The APADR was recently created by State Decree No. 27,695 of February 21, 2018, and aims to protect biological diversity, discipline the occupation process and ensure the sustainable use of the natural resources of the respective area.

Historically, although the Piranhas-Açu Estuary has suffered from some drastic river flooding in the last decade (2004, 2008 and 2009) as a result of extreme rainfall events (Medeiros and Zanella, 2019; Medeiros, 2019), it has been suffering with the effects of tidal flooding (type of focus flooding of this work). Recent reports, from the 2010s were found in local newspapers and blogs, as well as personal observation in loco (personal communication). Since then, due to the effects of the sea, streets and houses have been frequently invaded by water, bringing negative consequences to several communities of the northern coast of Rio Grande do Norte (Figure 2). The current facts, allied to the local disordered occupation, the environmental and economic importance of the region, the scenarios of SLR, and finally, the lack of literature on the case (tidal flood), make this region an area of great appeal for the development of scientific works to subsidize information for decision-making about climate change adaptations.

## 3 Material and methods

Tidal flood risk mapping was performed using data on the meteorological tide, astronomical tide and a high-resolution LiDAR Digital Elevation Model (DEM), both calibrated for the study area. The data were subjected to statistical analysis and return period calculations. The 20-year return period was adopted as the base reference quota for this study and allied with the projections found in the literature on the global SLR for the coming years. Finally, flood risk mapping was performed based on flood scenarios and the vulnerability of land use and cover.

### 3.1 Tide database

For this study, sea level variation was represented by the sum of the meteorological and astronomical tide (SMC-Brasil, 2018).

### 3.1.1 Meteorological tide (MT)

The MT or storm surge (also known as non-astronomical sea level) is the result of atmospheric forcing such as wind pressure or sea level pressure variations. To compose the local MT historical series, the maximum annual tide level was obtained from the data from point 19 (Lat. 4.821 ° W / Long. 36.500 ° S) (Figure 1) from the GOS (Global Ocean Surge) database of the SMC-Brasil project (SMC-Brasil, 2018). This database is a selection of regional reanalysis series, located along the Brazilian coast, built with a forced numerical simulation with atmospheric pressure fields and winds and validated for the region. To

generate the series, we used the barotropic module (2DH) of the model in a global mesh with 0.25° spatial resolution and bathymetry data from ETOPO2 model (NOAA). As oceanographic influences of the model, pressure data at sea level and global winds (10m high) from Reanalysis 1 (NCEP / NCAR) were used. The wind and pressure data have a spatial resolution of 1.9° and a 6-hour temporal resolution. Each of the series has duration of 60 years (1948-2008) with a time interval of 1 hour (SMC-Brasil, 2018).

### 3.1.2 Astronomical tide (AT)

AT is defined as the set of regular SLR and fall motions over 12 or 24 hours, produced by the gravitational effects of the Earth-Moon-Sun system. Other celestial bodies in the solar system are also exert gravitational force, yet weak when compared to and considered by the moon and sun. Tide description and prediction at a given location can be done by harmonic tidal analysis (Pugh, 1987). In this investigation, the Astronomical Tide data are the result of deterministic forecasts prepared and provided

by the Brazilian Navy's Directorate of Hydrography and Navigation (DHN). The maximum annual astronomical tide level was used from the astronomical tide forecast data released by the DHN, linked to the tidal gauge station of the Port of Macau (Lat. 4° 49' 05" S / Long. 37° 02' 04" W) (Figure 1), for the period 1998 to 2018. These data were available through tables showing the maximum and minimum daily values of astronomical tidal heights.

### 3.1.3 Statistical analyses of tide database

The Mann–Kendall sequential test (Mann, 1945; Kendall, 1975) was applied to evaluate the temporal serial behaviour of the annual maximum of meteorological and astronomical tides. The Mann–Kendall test is a robust, sequential, and non-parametric statistical method used to determine if a specific data series has a temporal tendency towards statistically significant changes. Among its advantages, it does not require normal distribution of data and is only slightly influenced by abrupt changes or non-homogenous series (Zhang et al., 2009). In recent years, with growing concerns over environmental degradation and about the

implications of greenhouse gases on the environment, researchers and practitioners have frequently applied the non-parametric Mann–Kendall test to detect trends in recorded hydrologic time series such as water quality, streamflow, and precipitation time series (Yue and Wang, 2004; Araújo et al., 2019). Although it has no influence on the tidal flood risk mapping, the Mann–Kendall test was applied to investigate if the elevation of tides is showing any upward or downward trend.

Subsequently, the data was submitted to fit the extreme value of the Gumbel distribution function (Gumbel, 1958). Extreme value statistics are used primarily to quantify the stochastic behaviour of a process at unusually large or small values. Particularly, such analyses usually require estimation of the probability of events that are more extreme than any previously observed. Many fields have begun to use extreme value theory, and some have been using it for a very long-time including meteorology, hydrology, finance and ocean wave modelling to name just a few (Gilleland and Katz, 2016). The Gumbel Distribution is also known as Type I extreme value distribution, or Fisher-Tippet type I distribution, and it has the function of accumulated probabilities given by Eq. (1):

$$F_X(x) = P\{X < x\} = e^{-e^{-y}}, \tag{1}$$

Being $x$ the ratio and $y$ the reduced Gumbel variable given by Eq. (2):

$$y = \frac{x - \beta}{\alpha}, \tag{2}$$

Where $\alpha$ and $\beta$ are characteristic parameters of the Gumbel line; $\alpha$ represents meter of scale and $\beta$ the position parameter. The return period (T$r$) in years can be obtained by Eq. (3):

$$x(Tr) = \beta - \alpha ln\left[-ln\left(1 - \frac{1}{Tr}\right)\right], \tag{3}$$

In this work, the 20-year return period (Tr20) was adopted as the starting reference for flood hazard mapping in question. All statistical analyses were performed using R software (R Development Core Team, 2020). The packages used were "Kendall" and "extRemes", for Mann – Kendall sequential test and to fit extreme value of Gumbel distribution function, respectively.

## 3.2 Adjustment of the Reduction Level to the Brazilian Geodetic System

The astronomical tidal information provided by the DHN is all linked to the so-called Reduction Level (RL), which is the altimetric reference system for bathymetric depth measurement adopted by the Brazilian Navy (MB). This reference system assigns the average low tides of spring to a measurement reference made at local level. Thus, the establishment of the local sea level found in the nautical charts and information provided by the Brazilian Navy has its own framework aimed at knowledge of the seabed relief for navigators' safety (CHM, 2019a, 2019b). Therefore, it is a different altimetric reference than the official geodetic reference system adopted by the country (Matos, 2005; Ramos & Krueger, 2009).

In order to standardize the altimetric reference of this work, Brazilian Vertical Datum of Imbituba was adopted in association with the Brazilian Geodetic System (SGB). For this purpose, a level reference point was traced over approximately 50 minutes employing a two-frequency GNSS receiver (L1 / L2) near the port of Macau (Figure 3). This landmark, codenamed RN-2 (DHN), was deployed during the construction of the respective navigational station by the Brazilian Navy. After the screening, the GNSS data were submitted to coordinate adjustment post-processing through MAPGEO2015 the official Brazilian Geoidal Undulation Model and its SGB-linked orthometric altitude was obtained through the Precise Point Positioning of the Brazilian Institute of Geography and Statistics (IBGE-PPP). IBGE-PPP is a free online service for GNSS (Global Navigation Satellite System) data post-processing that makes use of the GPS Precise Point Positioning (CSRS) program developed by the Geodetic Survey Division of Natural Resources of Canada. It allows users with GPS and/or GLONASS receivers to obtain coordinates

referenced to SIRGAS2000 (Geocentric Reference System for the Americas) and ITRF (International Terrestrial Reference Frame) through precise processing.

Finally, the reduction level (RL) orthometric altitude was obtained by mathematical subtraction operations at the RN-2 (DHN) level reference orthometric altitude, Eq. (4):

$$195 \quad H(RL) = H(RN) - 4.046m = 2.92m - 4.046m = -1.126m, \tag{4}$$

### 3.3 LiDAR Digital Elevation Model

An airborne LiDAR DEM was used, with 1m horizontal spatial resolution and coverage for the entire local area. The DEM was built by PETROBRAS, granted in terms of technical cooperation between the said institution and the Federal University of Rio Grande do Norte (UFRN) and made available by the Graduate Program in Geodynamics and Geophysics (PPGG), utilizing a confidentiality agreement. The survey took place between February and September 2012 through integrated aero photogrammetry. In this aerial survey, we used a model ALS60 equipment manufactured by Leica Geosystems. The average point density was 4.2 per m², with a frequency of 200,000 pulses per second (200 kHz) and a 26° aperture angle (FOV). The operating frequency adopted was 160.2 kHz, with 65 Hz profiling frequency and an average aircraft speed of 190 km/h. The average point spacing was 0.41 m (in the flight direction) and 0.80 m in the transverse flight direction. The geodetic reference system adopted for planimetric data was SIRGAS2000. For altimetry data, the geoidal undulation model MAPGEO2015 was used. The geometric altitude was converted to orthometric altitude to leave all in the same altimetry reference (all linked to the Brazilian Geodetic System). The altimetry RMSE of this product, obtained during the evaluation and calibration process (Araújo et al., 2018), was 0.1704 m.

### 3.4 Scenarios for mean sea level rise (MSLR)

Mean sea level rise (MSLR) has been widespread in the international community as one of the impacts related to climate change, where most estimates are projected by the year 2100. For this research, we adopted 3 scenarios of MSLR at global and regional scale to incorporate the predictions of average SLR until the year 2100. The forecast scenarios of the IPCC AR5 (Church et al., 2013), reviewed by Oppenheimer et al. (2019), using a collection of process-based models and supplementary database, projects a median and likely (66% probability) global MSLR of 0.53 m (0.36-0.71 m) and 0.74 m (0.52-0.98 m) by 2100 for RCP 4.5 and RCP 8.5, respectively. Also, stations of the Geodetic Permanent Tide Gauge Network (RMPG) operated by the Brazilian Institute of Geography and Statistics (IBGE) point out a MLSR of 2.1mm/year on Brazilian coast since 2001 (IBGE, 2016). RCPs are scenarios with a global projection, whereas the IBGE scenario is a projection at a regional level, mainly for the for Brazilian Northeaster coast. The IBGE scenario is the result from a simple linear projection based on data variation obtained by RMPG, while the IPCC scenarios are results from robust modelling of sea level projection in the face of climate change.

## 3.5 Land use and cover map

Land use and cover mapping of the study area was carried out in three stages arranged in a systematic way: 1) Digital Image Processing (DIP), mainly based on applications of images processing algorithms; 2) Fieldwork; 3) Cartographic digitization and manual vectorization.

The first stage consisted of highlighting thematic information using DIP techniques on multispectral optical images from the Operational Land Imager (OLI) sensor on the LANDSAT-8 platform, and Synthetic Aperture Radar (SAR) Multilook Fine image from the RADARSAT-2 mission. The DIP techniques included multispectral bands ratios and Principal Components Analysis (PCA) applied in the Red-Green-Blue (RGB) colour system and RGB-Intensity hybrid colour system. Thus, the multispectral colour composites R(7/5)G(6/4)B(5/4), R(PC5)G(PC6)B(PC7) and hybrid colour composites R(7/5)G(6/4)B(5/4)I(RADARSAT-2) were analysed in order to highlight thematic classes associated with the compositional differences of the soil, vegetation, water and anthropic materials. PCs 5, 6 and 7 showed a predominant contribution of 58% of band 6, 56% of band 4 and 60% of band 1, respectively, elucidating information from the middle, near and visible infrared spectrum. The use of RADARSAT-2 image sought to extract the maximum textural information from each target on the surface, due to the principles of diffuse and specular reflection of SAR images (Jensen, 2009).

The second stage consisted of *in loco* visit on the study area with the purpose of validate information extracted from DIP, in addition to providing analyses on a detailed scale. Correlations were made between field observations and land use and cover classes, recognized through spectral data on colour composite image.

Finally, in the third stage, all the information on the land use and cover mapping was integrated and mapped by manual vectorization on a scale of 1:10,000 with ArcMap 10 software.

## 3.6 Tidal Flood Risk Mapping

For this investigation, we adopted as a quantitative risk the likelihood of harmful consequences or expected losses (dead, injured, destroyed and damaged buildings, etc.) occurring as a result of interactions between a natural hazard and conditions of local vulnerability (UNDP, 2004). The formula proposed by Wisner et al. (2011), in which the same concept was one of the references of the fifth IPCC report (IPCC, 2014), Eq. (5):

$$Risk\ map = Hazard\ map\ X\ Vulnerability\ map, \tag{5}$$

Where, Hazard map is the likelihood of the process occurring with magnitude $M$ (destructive potential) and Vulnerability map (physical vulnerability) is the degree of damage or loss to the exposed environment as a result of the impact and as a function of magnitude $M$.

For tidal flood hazard mapping, was assigned 4 classes based on the scenarios under study, in addition to the current flooding (Table 1). Each class represented the quota resulting from the sum of:

+ Projection of MSL elevation to 2100;

+ Meteorological tide (Tr20);

+ Astronomical tide linked to SGB (Tr20);

+ RMSE of DEM.

For the construction of the flood vulnerability mapping, was used as the basis the land use land cover mapping previously designed for this work. The land use and cover mapping vector file was transformed into a raster file, with a spatial resolution of 1m due to the spatial resolution of the altimetry data (DEM data). Finally, the raster file was reclassified with vulnerability values, in scores from 0 (no vulnerable) to 5 (most vulnerable), assigned to the land use and cover category (Table 2).

After obtaining the flood hazard and vulnerability maps, the risk map was obtained using the risk equation mentioned above. The risk was classified into 5 cassettes according to the values in Table 3.

## 4 Results and discussion

The flood risk mapping from high-resolution DEM provides the knowledge to optimize investments and provide flood risk management with high accuracy (Schröter et al., 2018). The standardized altimetry reference in centimetre intervals become

essential, especially when there is an interest in the analysis of land use and cover, or when economic activities occur in these environments in order to estimate risks with accuracy (Aguiar et al., 2019).

The flood quota reached in a region is a particularly complex phenomenon, both in the number of elements involved in the flooding process and in the interaction between these elements. However, it was possible to robustly model the complexity of tidal flooding that occurred in the Piranhas-Açu Estuary, as well as the risk in its probabilistic potential for the coming years.

**4.1 Tidal behaviour and return period**

The effect of AT and MT on the coast is observed as a variation of sea level or free surface, and it is at this level that waves propagate (SMC-Brasil, 2018). From the GOS data and the DHN data applied in this research, it was possible to observe the tidal behaviour in the tropical Pianco-Piranhas-Açu estuary (Figure 4).

The meteorological surge or storm surge is a sea level fluctuation caused by weather effects mainly derived from wind and

variations in pressure fields. Throughout the 61 years of data from the GOS point, the maximum annual quota of the meteorological tide presented an average of 12cm and an amplitude of 14cm, with a maximum of 22cm and a minimum of 8cm in the years 1964 and 1958, respectively. When applying the Mann-Kendall test, no statistically significant trend (Tau = -0.123; p= 0.16147) was observed in the dataset.

The astronomical tide is the result of the interaction of the gravitational forces of Earth, Moon and the Sun, being completely

predictable. With the DHN data set provided by the Brazilian Navy, it was observed that the maximum annual astronomical tide quota presented an average of 2.80 m and a 13 cm amplitude, with a maximum of 2.84 m and a minimum of 2.71 m. In the same study, when the Mann-Kendall test was applied, no statistically significant trend was observed (Tau = 0; $p$= 1), showing a steady pattern.

The descriptive values on the tides presented to corroborate the values found in the literature. Frota et al. (2016), studying the
tidal behaviour in the Brazilian Northeast during the period from 2009 to 2011 in buoys about 200 km from the Piranhas-Açu estuary, found that the average maximum tide height was 2.79 m, ranging from 2.23 to 3.34 m. In the same study, Frota et al. (2016) found that the sea level variability in the sub-FT (The non-astronomical sea level signal) represents low oscillation, with a maximum of 0.12 m. Mattos et al. (2019) made a scientific expedition from December 2010 to February 2011 to study significant wave heights and found that the tide table of Guamare-RN (approximately 40km east of the table of Macau-RN)
had averages of 2.34 m (in syzygy tides) and 2.21 m (in quadrature tides), both above the reduction level.

Regarding the return period (Tr20years), estimated by the Gumbel distribution function, the values of 15.90 cm and 2.90 m were found for MT and AT, respectively (Figure 5).

By performing the geodetic tracking of the DHN RN-2 framework, Macau´s tide gauge was adjusted to the Brazilian Geodetic System (SGB) and in it, we found the Reduction Level (RL) orthometric altitude, represented by the value of -1.126 m (Figure
3). Thus, the orthometric altitude of the maximum astronomical tide quota for the 20-year return period was 1.777 m, which served as a start for flood models.

**4.2 Tidal flood hazard and vulnerability maps**

It was possible to produce the four classes tidal flood hazard map for the study area based on the mean sea level projection values for the year 2100 (Table 5.4). In this map, the use of the astronomical tide quota in association with the Brazilian
Geodetic System (SGB) was of paramount importance, thus ensuring that all input variables for flood hazard mapping were in the same geodetic framework.

After the spatialization of the classes in a GIS environment, we verified the spatial behaviour of the tidal flood hazard throughout the Piranhas-Açu estuary (Figure 6a). In general, there was a positive north-south gradient, with a predominance of the flood class of the present scenario (high hazard). The high hazard class represented 257.60 km² of the estuary flood
hazard, while the moderate hazard, low hazard, and extremely low hazard classes represented 286.26, 338.67 and 359.42 km², respectively.

Tidal flood stains were observed inside the urban area of the city of Macau. These spots are justified by the current layout of the city's drainage system, where at high tide times sea water enters the galleries and canals, affecting the interior of the city (Figure 6b and Figure 7). Aguiar et al. (2019) found the same structural problem in the urban area of the city Areia Branca
(approximately 58 km west of the city of Macau). It is important to mention that the land on which the local cemetery is in the urban area of Macau is one of the few urban sectors in the city not to suffering from tidal flood scenarios. This result becomes extremely important for the future urban planning of the city.

It was found that the flood event of January 3, 2015, had an orthometric altitude of 1.73m (Figure 8). The same spatial pattern of tidal flooding was observed between the photographic record and the flood model proposed in this work, thus, validating
the applied flood model.

By ranking the mapping of the land use and land cover, we obtained the quantification of the areas of the mapped units (Table 5 and Figure 9), highlighting the Caatinga area, which corresponded to 657.18 km².

Regarding the vulnerability map, it was observed that 66.86% (883.84km²) of the vulnerable areas had low flood vulnerability (Table 5.5 and Figure 10). However, it is important to note that 16% of vulnerable areas have high and extremely high vulnerability, corresponding to an area of 205.30 km².

### 4.3 Tidal flood risk map

From the result of flood hazard and vulnerability mappings, the flood risk map was obtained (Figure 11).

The risk areas represent a total of approximately 360km², where the 135.23km² low-risk class stands out, while the other classes represented 85.64km² (extremely low risk), 20.25km² (moderate risk), 117.73km² (high risk) and 0.53km² (extremely high-risk). Extremely high-risk environments were sections of the urban areas of the cities of Porto do Mangue and Macau, and the communities of Ponta do Mel, Rosado and Diogo Lopes (Figure 10).

### 5 Conclusions

The SLR by a few millimetres per year is an important variable since loss of land in lowland areas can quickly destroy coastal ecosystems such as lagoons, coastal lakes and mangroves. In addition to flooding of socio-economically and environmentally sensitive relevant areas, the SLR can change the energy balance of coastal environments, causing large variations in the sedimentary process and, consequently, erosion of large stretches of shoreline (Castro et al., 2010). In the Piranhas-Açu Estuary, SLR was not statistically significant, we believe that the temporal scale of meteorological tide data set (1948 to 2008) favoured the masking of this phenomenon Since it reported by the local community and the news for the last 10 years only.

It is possible that tidal flooding in the region under study is closely linked to rising sea levels in recent years. Extreme tidal weather events are the main factor in flood danger. Flood hazard, vulnerability and risk maps are crucial for planning and intervention in flood prone areas. The case study results in the Piranhas-Açu Estuary can be used by local environmental management mainly to characterize risk zones and support the implementation of tidal flood risk management plans in this coastal area. The materials and methodology applied to this study area have proven to be effective in identifying tidal flood risk areas using high-resolution DEM that has been calibrated based on high precision GNSS, historical tidal quota data and geoprocessing techniques.

It is noteworthy that the methodological approach to the Piranhas-Açu Estuary is suitable to be replicated with other estuaries, particularly those in Brazilian semi-arid regions (estuaries with low hydrological contribution from rivers). The application of tidal flood risk mapping may be particularly useful for regions with a good historical series of tidal database. In this case study, the tide flood event modelling of 2015 was compared with the photographic records of the respective event and established high visual similarity between them.

This paper also demonstrates that well-applied geoprocessing techniques, such as GIS and high precision geodetic features, provide results that can be very effective in environmental management with low-cost investments, highlighting the unique features of a given locality, especially floodplains and wetlands.

**Conflicts of interest**

None.

**Acknowledgments**

The authors express special thanks to PETROBRAS for providing LiDAR Digital Elevation Model data. We also thank the reviewers of the Natural Hazards and Earth System Sciences (NHESS) journal for their many insightful comments.

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

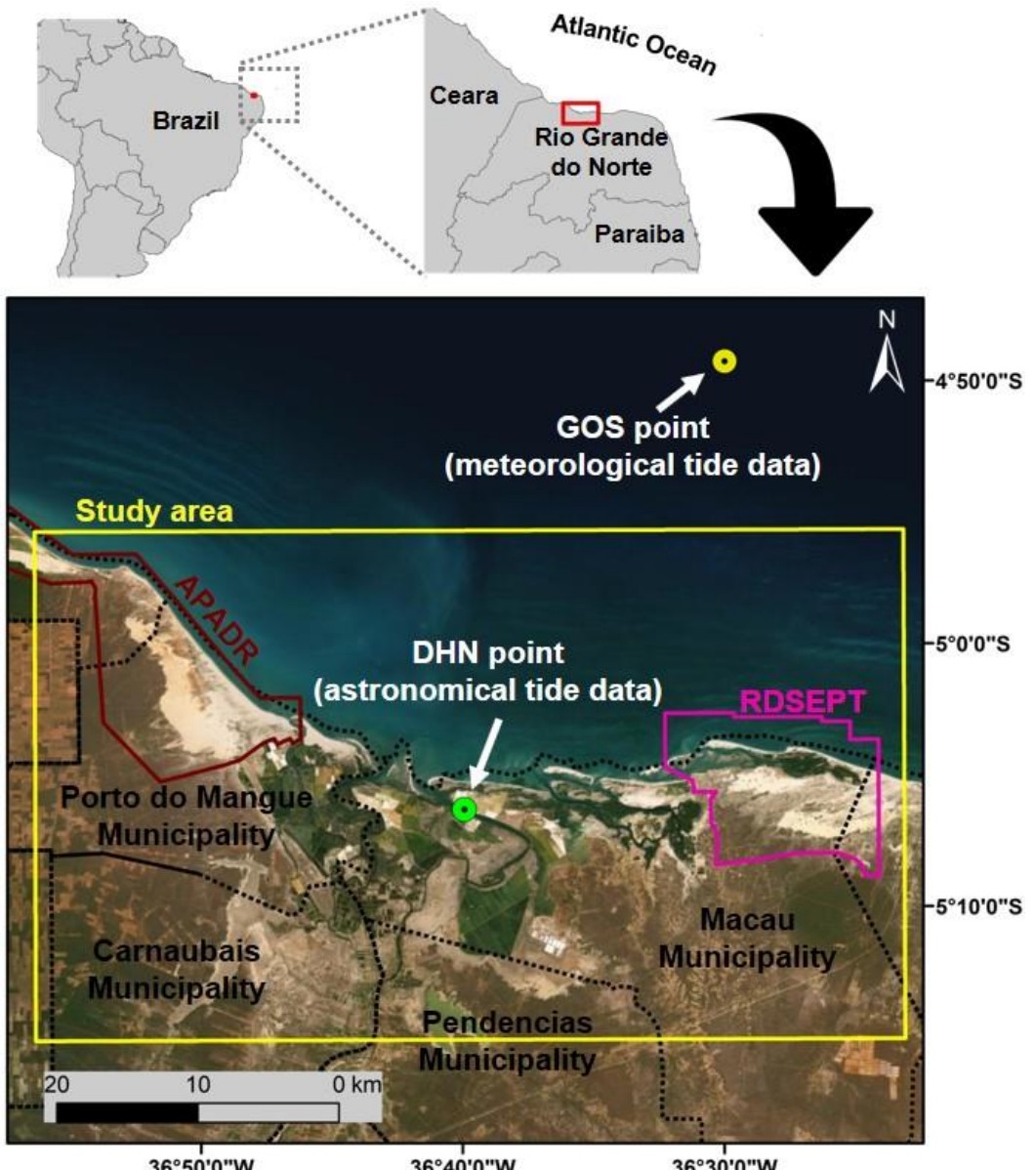

**Figure 1: Location of the study area. The yellow rectangle represents the delimitation of the study area: Pianco-Piranhas-Açu estuary and its surroundings. The pink polygon highlights the boundary of the Ponta do Tubarao State Sustainable Development Reserve (RDSEPT), while the burgundy polygon delimits the Rosado Dunes Environmental Preservation Area (APADR). The black dotted line represents the geopolitical boundaries of the municipalities that make up the study area. Basemap from ArcGIS Online: © ESRI.**

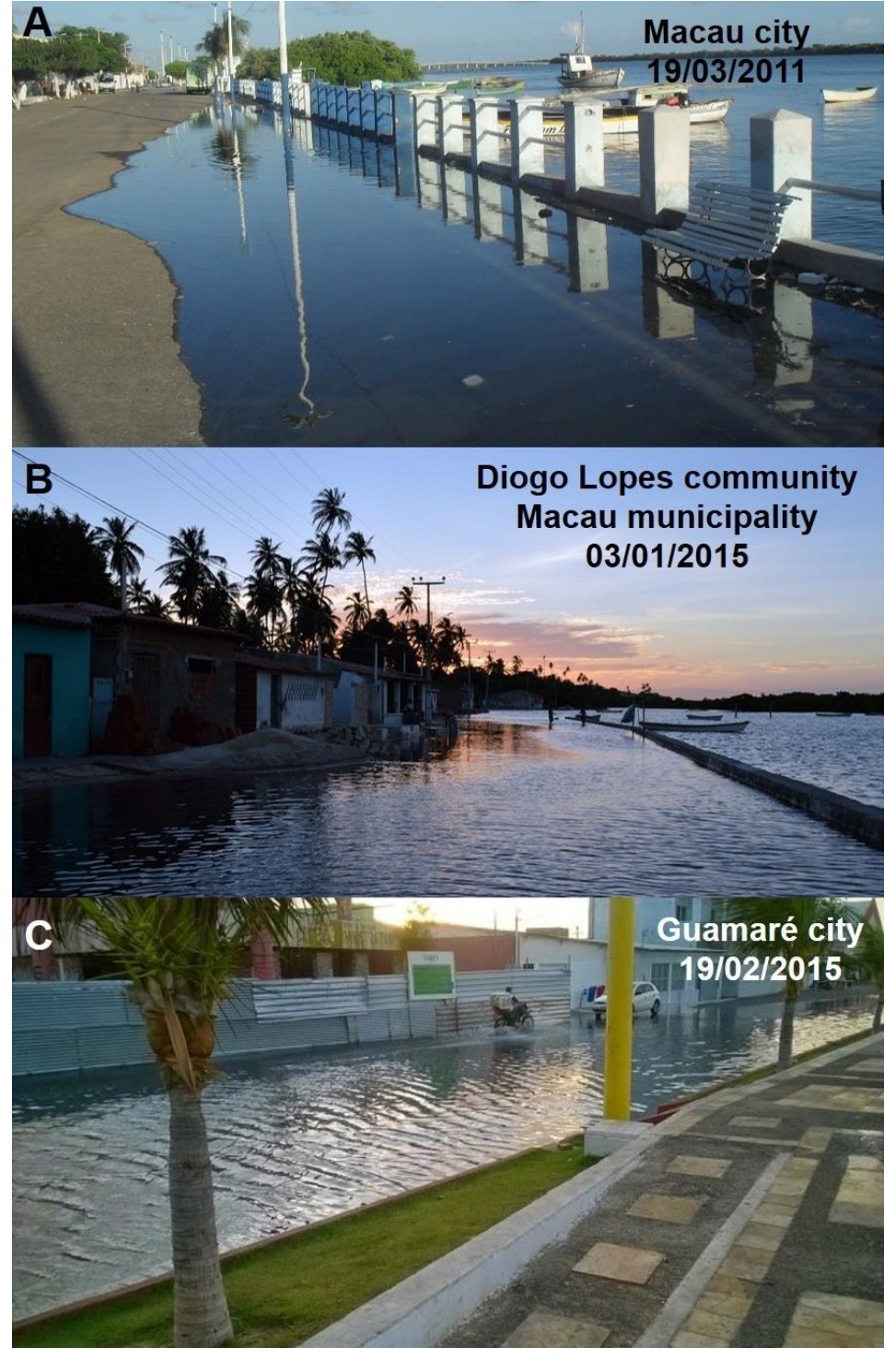

**Figure 2: Tidal flooding events in north coastal of Rio Grande do Norte State. (A) Macau urban area, Macau municipality, Mar 19, 2011 (Unknow author); (B) Diogo Lopes community, Macau municipality, Jan 03, 2015 (Tiago Ezequiel); and (C) Guamare urban area, Guamare municipality, Feb 19, 2015 (Unknown author).**

545

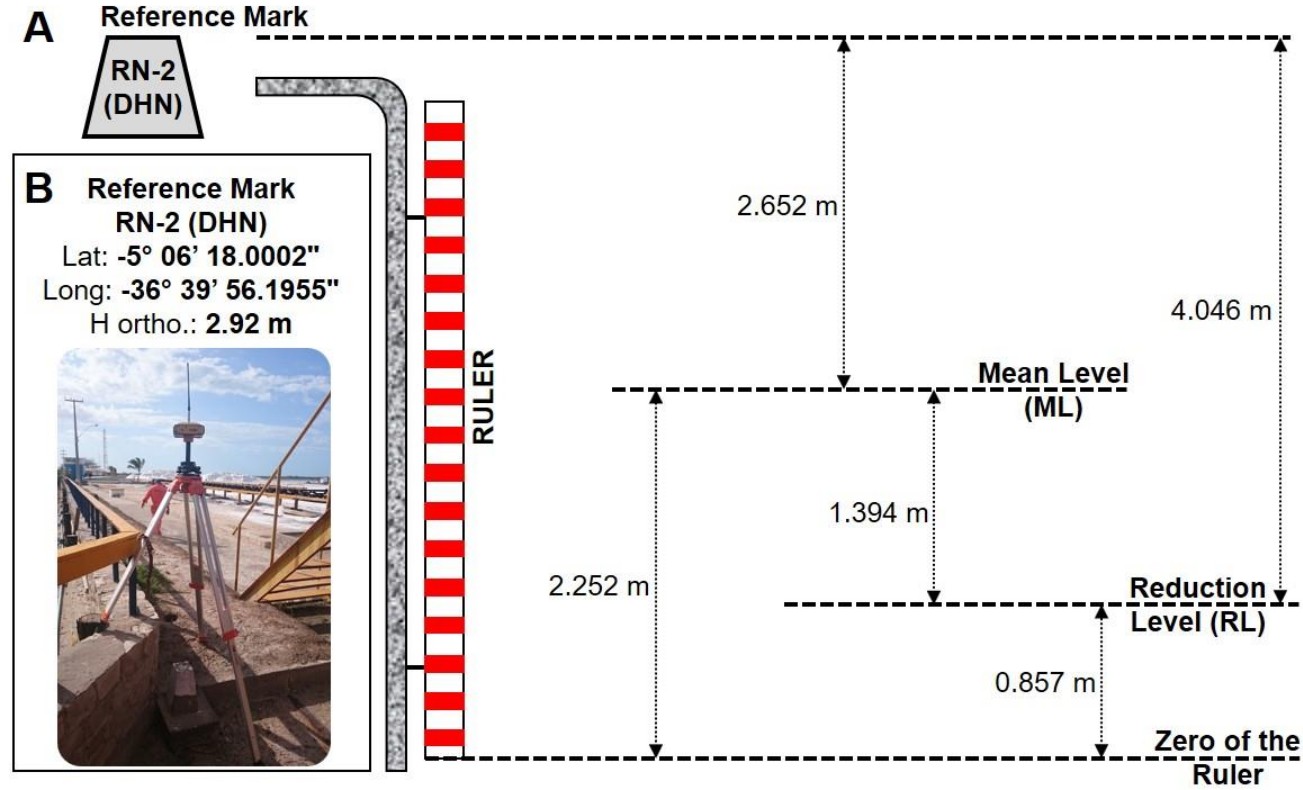

**Figure 3: Scheme illustrating the differences in quotas between the references adopted by the Brazilian Navy: (A) Reference Mark RN-2 (DHN); and (B) Coordinates of Reference Mark RN2- (DHN).**

550

555

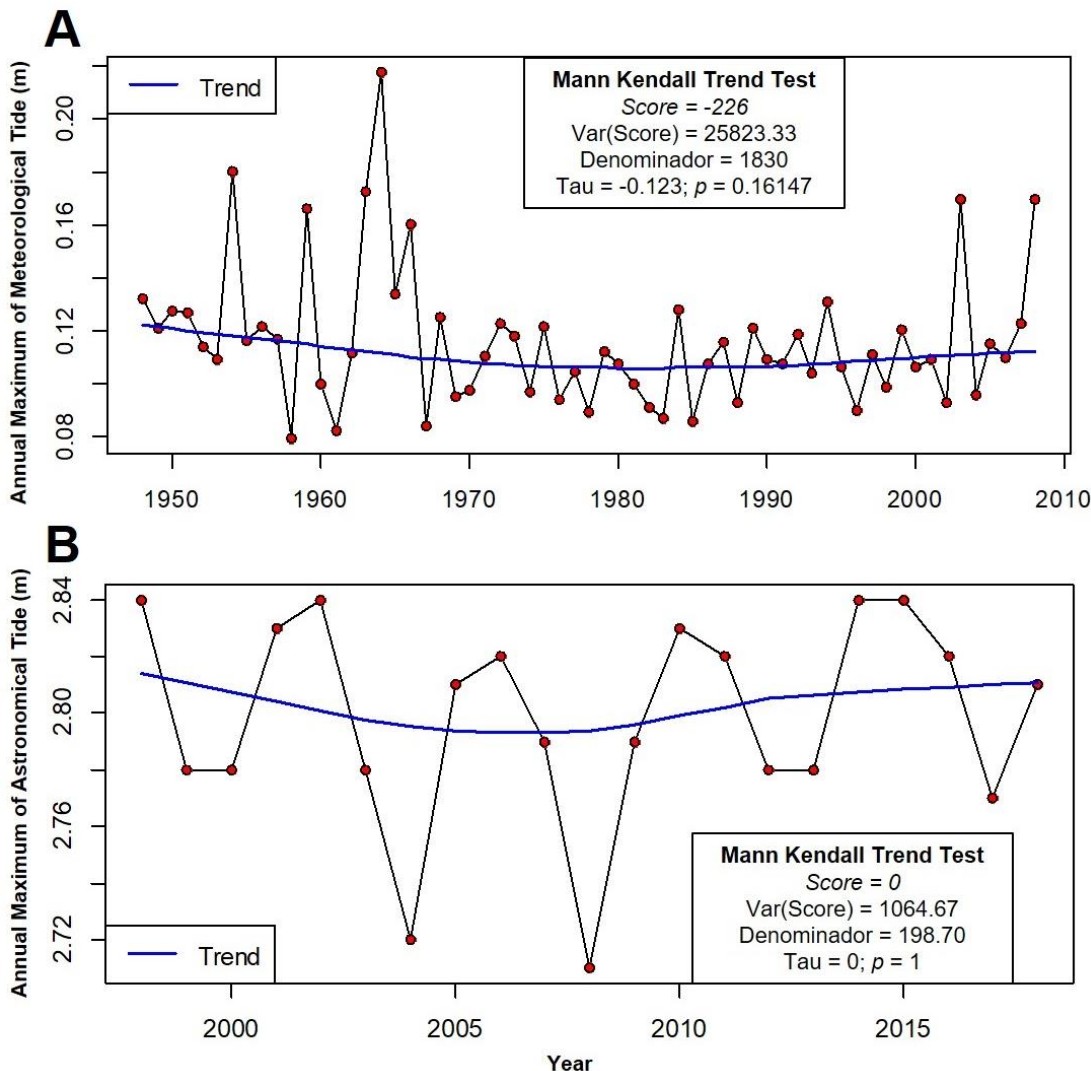

**Figure 4: Annual tide maximum level temporal serie: (A) Meteorological tide; and (B) Astronomical tide.**

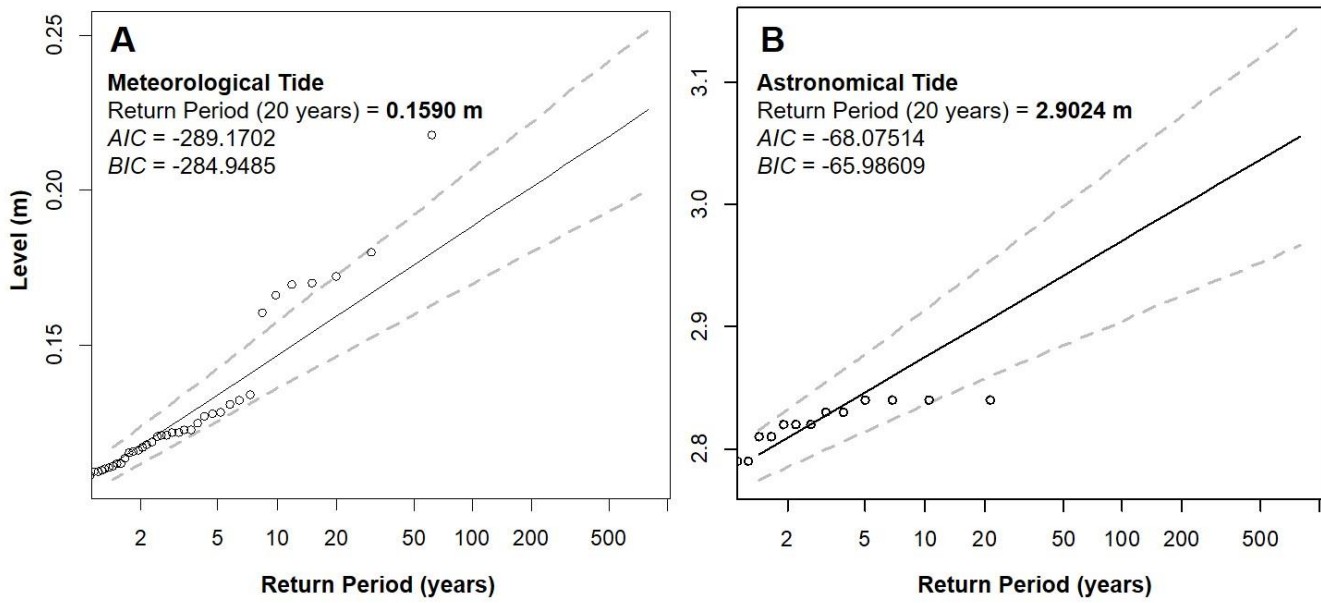

**Figure 5: Return period graph of tide data: (A) Meteorological tide; and (B) Astronomical tide.**

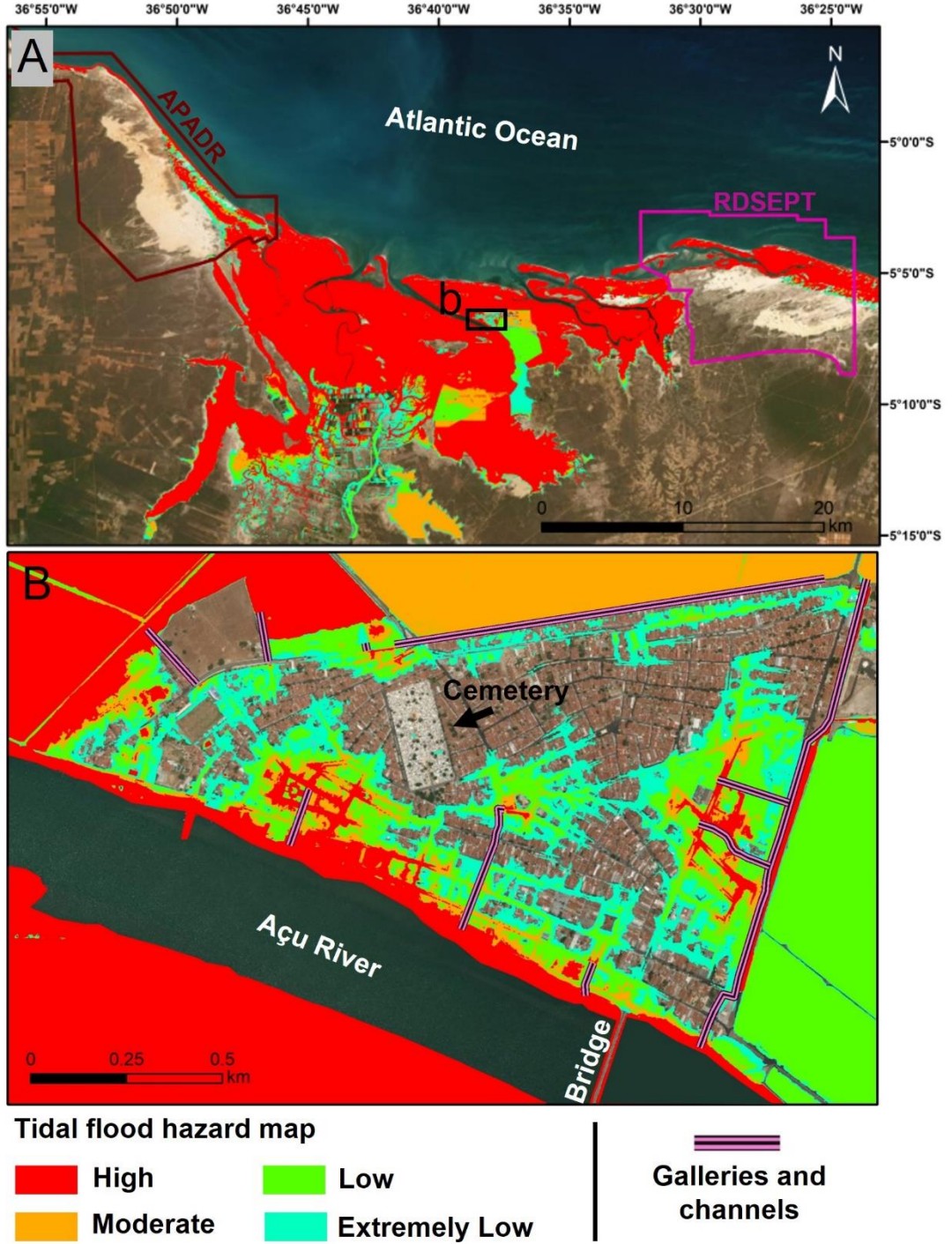

**Figure 6: Tidal flood hazard map: (A) Total area under study; and (B) Detail in Macau urban area. Basemap from ArcGIS Online: © ESRI.**

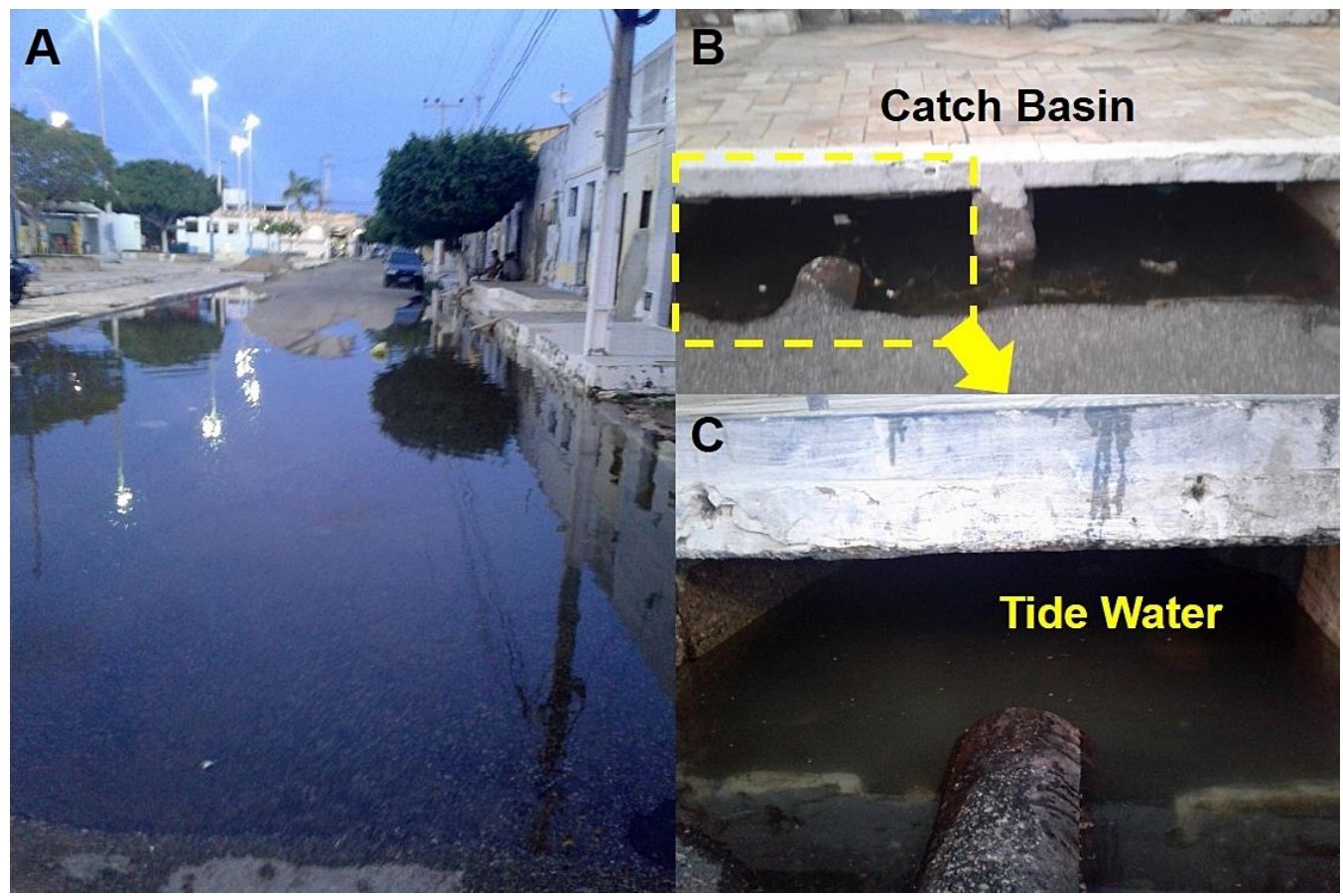

**Figure 7: Tidal flood through the rain drainage system: (A) Tidal flooding event on March 10, 2020 (Macau urban area); (B) Example of local catch basin; and (C) Catch basin in detail.**

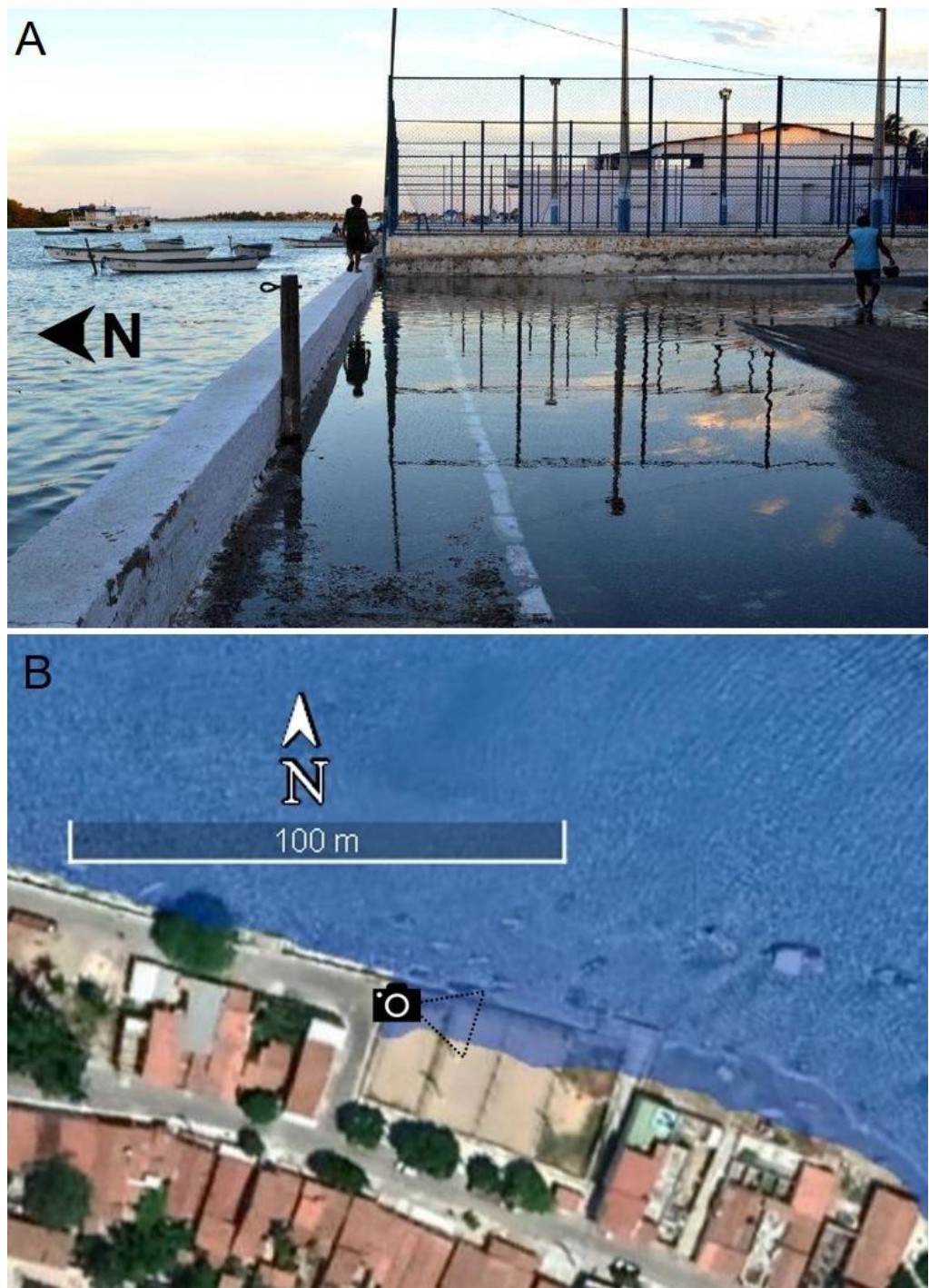

**Figure 8: Tidal flooding events in north coastal of Rio Grande do Norte State: (A) Photographic record of Diogo Lopes community, Macau City, on January 3, 2015. (Tiago Ezequiel); and (B) simulated flood event for Diogo Lopes community on January 3, 2015. (orthometric height of tidal flood = 1.73m) (Basemap from Google Earth Pro: © Google LLC).**


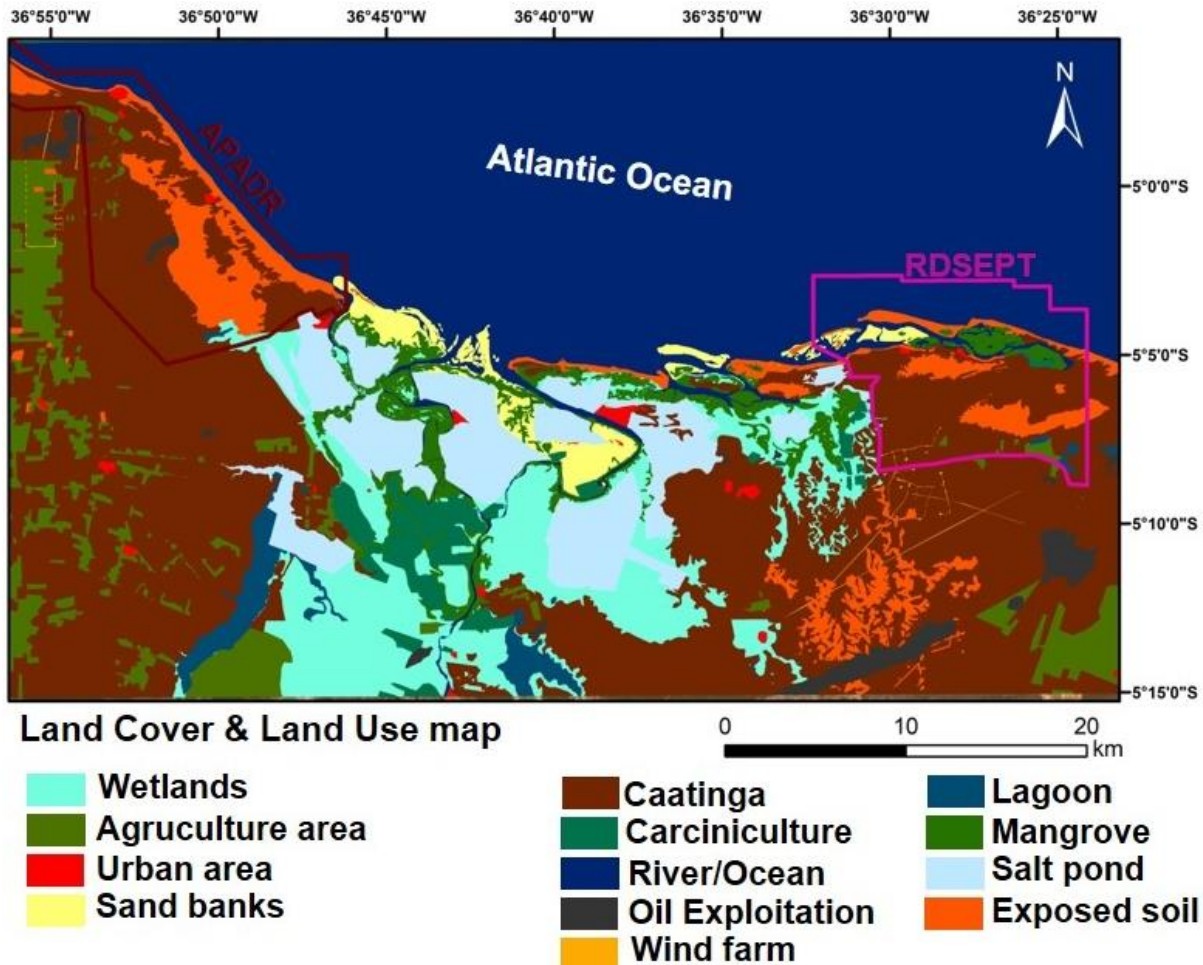


**Figure 9: Land cover and land use map for study area.**



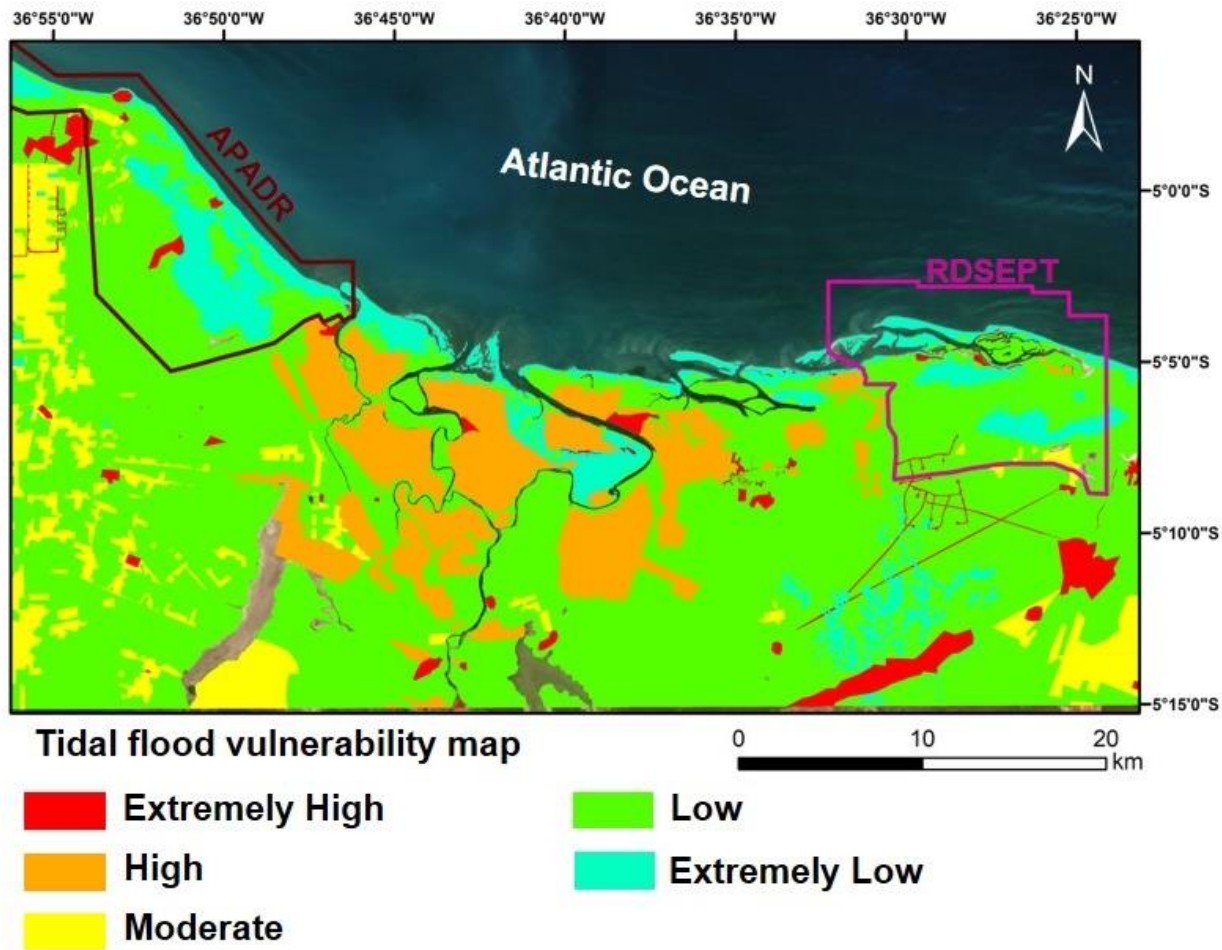

**Figure 10: Tidal flood vulnerability map.**



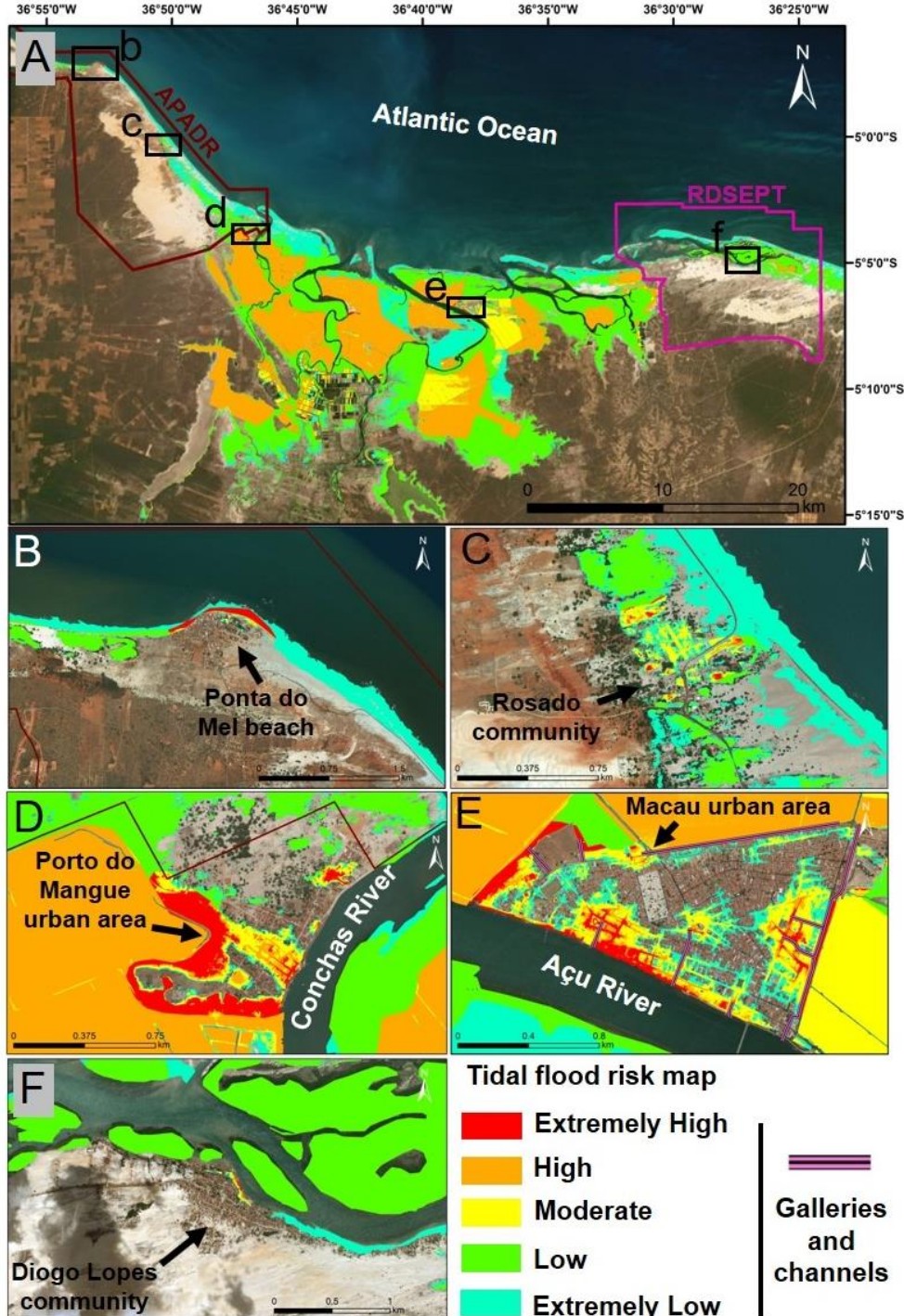

**Figure 11: Tidal flood risk map for Piranhas Açu Estuary: (A) View of entire study area; (B) Ponta do Mel community; (C) Rosado community; (D) Porto do Mangue urban area; (E) Macau urban area; and (F) Diogo Lopes community. Basemap from ArcGIS Online: © ESRI.**

**Table 1: Attributed value to hazard map.**

| Scenario | Value | Hazard |
|----------|-------|--------|
| Present | 5 | High |
| IBGE | 4 | Moderate |
| IPCC RCP 4.5 | 3 | Low |
| IPCC RCP 8.5 | 1 | Extremely Low |

**Table 2: Attributed value to vulnerability map.**

| Category | Value | Vulnerability |
|----------|-------|---------------|
| Urban area | 5 | Extremely High |
| Oil Explotation | 5 | Extremely High |
| Wind farm | 5 | Extremely High |
| Shrimp farm | 4 | High |
| Salt pond | 4 | High |
| Agriculture area | 3 | Moderate |
| Wetlands | 2 | Low |
| Caatinga | 2 | Low |
| Mangrove | 2 | Low |
| Sand banks | 1 | Extremely Low |
| Exposed soil | 1 | Extremely Low |
| Lagoon | 0 | No vulnerability |
| River/Ocean | 0 | No vulnerability |

**Table 3: Risk map value range.**

| Risk | Value Range |
|------|-------------|
| Extremely High | $> 20$ and $\leq 25$ |
| High | $> 15$ and $\leq 20$ |
| Moderate | $> 10$ and $\leq 15$ |
| Low | $> 5$ and $\leq 10$ |
| Extremely Low | $> 0$ and $\leq 5$ |


**Table 4: Tidal flood quotas in the scenarios under study.**

| Hazard Class | Projection of MSL elevation to 2100 (m) | Meteorological tide (Tr20) (m) | Astronomical tide linked to SGB (Tr20) (m) | RMSE of DEM (m) | Flood quota (m) |
|---|---|---|---|---|---|
| High | --- | 0.1590 | 1.7764 | 0.1704 | **2.1058** |
| Moderate | 0.1764 | 0.1590 | 1.7764 | 0.1704 | **2.2822** |
| Low | 0.5300 | 0.1590 | 1.7764 | 0.1704 | **2.6358** |
| Extremely Low | 0.7400 | 0.1590 | 1.7764 | 0.1704 | **2.8458** |


**Table 5: Area of land cover and land use categories.**

| Category | Area (km²) | Tidal Flood Vulnerability | Area (km²) |
|---|---|---|---|
| Urban area | 7.84 | | |
| Oil Explotation | 27.37 | Extremely High | 40.17 |
| Wind farm | 4.96 | | |
| Shrimp farm | 34.19 | High | 165.13 |
| Salt pond | 130.94 | | |
| Agriculture area | 115.10 | Moderate | 115.10 |
| Wetlands | 174.74 | | |
| Caatinga | 657.18 | Low | 883.84 |
| Mangrove | 51.91 | | |
| Sand banks | 29.86 | Extremely Low | 117.61 |
| Exposed soil | 87.75 | | |
| Lagoon | 24.22 | No vulnerability | 873.20 |
| River/Ocean | 848.97 | | |


**Table 6: Area of Tidal Flood Risk in study area.**

| Tidal Flood Risk | Area (km²) |
|---|---|
| Extremely High | 0.53 |
| High | 117.76 |
| Moderate | 20.25 |
| Low | 135.23 |
| Extremely Low | 53.83 |
| Total | 327.60 |