# Peer review of "Tidal flood area mapping fronts the climate change scenarios: case study in a tropical estuary of Brazilian semiarid region"

_Natural Hazards and Earth System Sciences, 2020_

## Referee Comment (RC1) · Anonymous Referee #1 · 13 May 2020

The paper is relevant to the scope of the journal. It a case-specific article and author(s) has presented the case of Rio Grande do Norte Brazil which was affected by Tidal flooding.

Author(s) attempted to prepare a Flood Hazard, Vulnerability and Risk map based on a method proposed by Wisnert et.al (2011). MT, AT, and high-resolution DEM is considered for preparation of different scenario. The different land area under risk is quantified and future management measures have been suggested.

Step by sept comments are cited here to improve the quality of paper

Title:

[Figure]

In the title, after the semiarid, add the "Region".

Abstract: Line 20, approximately 118.26 km2; this value is not matched with the values included in the table. Verify and include the correct value.

Introduction: The literature lacks recent advances in tidal flood mapping case studies. The application of modeling techniques for tidal flood mapping and remedial measures cases worldwide needs to include to strength the introduction part.

Study area:

What is the significance of Line 64-71 in tidal flood analysis? Line 88-90, presents about the mosaic study area, however, the area is not presented or marked in any figures.

Line 102-103, presented that the study area is also affected by river flooding, and however in the present case described only tidal flooding. Under this condition presented result for tidal flooding deficit the inundation case by river flooding. It would be considered for precise flood risk mapping besides tidal flooding or it would justify not considered river flood in the present case.

Materials and methods:

Description and preparation of land use land cover map for vulnerability mapping are missing.

Line 120, data from point 19, however, the figure 19 is not presented in figure 1, in addition, Legend is missing in Figure 1.

Line 179, LiDAR DEM covers an entire area, however, the area covered in km2 and in form of a map is missing in presented research.

Table 1, Hazard attribute values, in score 1 to 5, however score for 2 is missing. Justify why it is not considered. Table 2, Vulnerability values, in score 1 to 5, however, the score for 0 is missing. Justify why it is not considered or line 216 Score from 1 instead

it will start from Score 0.

Results and Discussion:

What is the significance of the use of High-resolution DEM (LiDAR) data in this work? Why the flood depth map is not presented in this study?

Flood waves are dynamic quantities, however, the entire concept presented in static condition. If dynamic tidal wave model or unsteady tidal flood modeling will be performed considering the river flooding and rain flooding will generate the different flood inundation or risk scenario.

Line 269, Table 5.5, However it does not match with the table numbers

References:

Line 309, reference 1 should cite in the English language.

Page 17, Photographs A to C, shows the tidal flooding in different years i.e. 2011, 2015, what is the relation with the scenario generated. How it will be utilized for validation, although the scenario is generated based on the 20 year return period

These are the major deficiencies observed in this work and need to improve.

I rated this paper as a major revision; Happy to review a revised version with significant improvement.
* * *

---

## Referee Comment (RC2) · Anonymous Referee #2 · 21 May 2020

General comments: I have just finished reviewing the manuscript entitles 'Tidal flood area mapping fronts the climate change scenarios: case study in a tropical estuary of Brazilian semiarid' by Araújo et al. Overall the deals with a very interest subject in an area with little published information. I believe that the article is of interest and within the scope of the journal but in my opinion needs significant improvements before publishing both methodological and scientific (see detailed comments). The most important of the problems is that the hazard methodology and classification is not presented correctly (looks confusing) and it needs further improvements. The data and methods used are presented in a way that is not easy to judge is the analysis used is correct. Abstract and Introduction sections very general and they do not help the reader to understand

the processes related. Finally, the objective presented at the end othe Introducion are rather technical. Study area has a lot of information that is not necessary or related with the objectives of the article an results and discussion need further work.

Specific comments:

Abstract 'Previous studies on tidal flood mapping are mostly with continental and/or global scale approacher' this statement is not correct. There are far more local scale flooding studies, it is true that the impact of the global or continental wide studies tends to be higher but the number of studies is not. Introduction LINE 26: 'has occurred at an accelerating rate' is occurring, as it continues to occur.

LINES 33-34: 'Decades ago, the flooding usually happened only during a powerful or localized storm now can happen when a steady breeze or a change in coastal current overlaps with high tide (NOAA, 2019).' This statement is only valid for the US area and for 'sunny day flooding' no storm contitions. It cannot be extrapolated to other areas since local environmental and infrastructure (drainage system) parameters are important. Please rephrase. Also explain in the introduction with what type of flooding this work is focused marine flooding, flooding through the drainage system or both.

LINE 35: 'In Brazil, the current panorama of coastal flooding is extremely worrying (Losada et al., 2013).' In what terms, this statement is not supported by the paper. Please explain which is the worring factors according to Losada etal. Since this is a local study please provide information related with the specific study area.

LINE 55: the objective as it expressed looks more technical than scientific. Please try to reformulate it.

Study Site

The first and the third paragraph have a lot of repeated information (in the first one without proper referencing). Please remove.

LINES 82-87: The tidal information is not properly provided. What is the mean tidal

range, the mean high water spring tide and the mean high water neap tides? The reduction level (RT) is a national level that is not explained until the methods section. Please refer ranges relative to MSL

LINE 103: 'been suffering lately' what do you mean by lately? Since when? How many events per year?

LINE 116: It is not possible to have a one line section. Please provide more information on the tidal data. Length of the record, precise location, gaps in the record frequency of data recording, reference level and data treatment. LINE 120: provide model resolution and other models parameters. Any indicators on the ability of this model to predict correctly the observed storm surge? Please provide calibration and validation information.

LINE 124: No need to define the Astronomical tide

LINE 130: It is not clear if the data are measured or predicted water levels. If the data are max and min measured then they are not purely astronomical data but they have also meteorological and judging by the figure 1also river components. If on the other hand the data are predicted max. and min. values from tidal constituents then there is no reason to calculate the return period of the data there is no probabilistic part only deterministic. Please describe better the data and adjust the statistical methods used in the text.

LINE 141: see comment above

LINE 153: change 'payback' with 'return'

LINE 195: 'Church et al., 2013)' not in the reference list. Also is this a global MSL estimation? Since it is a local study a local estimate would be better. You do not explain where you use the climate change data. Their use in Table 1 is confusing since extreme climate change scenarios are related with low hazard level. In general the use of CC scenarios in Hazard levels is confusing.

LINE 205-210: It is no clear on how you select your hazard levels and also which is you hazard indicator (see Ferreria et al., 2017 Process-based indicators to assess storm induced coastal hazards. Earth-Science Reviews 173). It is always better to select a proxy that is a measure of hazard (e.g. inundation depth instead of total water level.). Table 1 is not easy to understand as RCP8.5 is a low hazard scenario. In general this section is not clear and needs more work. Normally hazard can take a value of 0 'no hazard'.

LINE 212: The vulnerability is actually land use vulnerability. Which were the parameters used for the classification. Economic importance or other? The classification was based of regional or national stakeholders or was made by the authors?

LINES 230-237: It is not easy to observe trend with relative short time-series. Have you checked if there is a correlation with any of the large weather patterns and indices that affect the area? It is important to explain what kind of tidal data you present see my comment in methods section. Is it possible to extend the surge and tidal data to cover the events you have measurements?

LINES 240-247: this information is better in the study area.

LINES 251_301: it is difficult to comment on the results since I the hazard classification is not clear. Up to the results you do not mention which is the process of inundation (flooding form the see or drainage water). Please introduce this earlier in the text with the appropriate information of the type and location of the drainage system. Also how you do the mapping in GIS? Are you using an algorithm with hydraulic connectivity or it is a bathtub model?

---

## Author Comment (AC1) · 10 Jun 2020

Dear Referee #1,

We do appreciate your constructive, thoughtful, careful, and helpful comments and suggestions. After careful discussions and analyses, we finished the preparation of responses to you. If there are any new comments or suggestions, please let us know.

In this document, we respond to the comments received point by point.

Best Regards,
Paulo Victor N Araújo and coauthors

**Response to General Comment:**
**Referee #1:** The paper is relevant to the scope of the journal. It a case-specific article and author(s) has presented the case of Rio Grande do Norte Brazil which was affected by Tidal flooding.
**Authors' reply:** Many thanks for these kind words, we are pleased that the reviewer finds the manuscript to be interesting, relevant and well written.

**Referee #1:** Title: In the title, after the semiarid, add the "Region".
**Authors' reply:** We accepted and modified in manuscript.

**Referee #1:** Abstract: Line 20, approximately 118.26 km²; this value is not matched with the values included in the table. Verify and include the correct value.
**Authors' reply:** The value is correct. Maybe, due to lack of a table with "tidal flood risk values", Referee # 1 was confused with values present in tidal flood vulnerability table (table 5). In order to avoid future mistakes, we added in new version of manuscript a specific table with the tidal flood risk values (table 6).

**Referee #1:** Introduction: The literature lacks recent advances in tidal flood mapping case studies. The application of modeling techniques for tidal flood mapping and remedial measures cases worldwide needs to include to strength the introduction part.
**Authors' reply:** We agree with Referee # 1. We enriched the introduction with mentioned points in new version of manuscript.

**Referee #1:** Study area: What is the significance of Line 64-71 in tidal flood analysis?
**Authors' reply:** We understand that it would be pertinent to raise a brief summary of basin under study, in order to contribute to other studies. But we remove these lines and leave this description for another manuscript.

**Referee #1:** Line 88-90, presents about the mosaic study area, however, the area is not presented or marked in any figures.

**Authors' reply:** In fact, we do not present the features in any figure, but we believe it is extremely important to keep the lines in manuscript, as a way of describing and contributing to study area.

**Referee #1:** Line 102-103, presented that the study area is also affected by river flooding, and however in the present case described only tidal flooding. Under this condition presented result for tidal flooding deficit the inundation case by river flooding. It would be considered for precise flood risk mapping besides tidal flooding or it would justify not considered river flood in the present case.

**Authors' reply:** Although "tidal flooding" is the focus of the work, we believe it is also important to mention that the region has suffered some "river flooding" in the past. It is also important to emphasize that the main river is barred in several stretches and that, in studied section, river flooding only occurs in the special rain conditions. The river flooding in region were caused by the accumulation of water from extreme rainfall events in the hydrographic basin. However, after 2010's, tidal flood events have frequently occurred. This factor, together with the lack of literature on the case, corroborates it immensely for publication of this case study.

To improve the understanding of manuscript, we add in lines 104 and 106 the following sentence: "(type of flooding focus on this paper)" … "and finally, the lack of literature on the case (tidal flood)".

**Referee #1:** Materials and methods: Description and preparation of land use land cover map for vulnerability mapping are missing.

**Authors' reply:** Agreed with Referee # 1. We added a new topic in line 207: "3.5 Land use and cover map".

**Referee #1:** Line 120, data from point 19, however, the figure 19 is not presented in figure 1, in addition, Legend is missing in Figure 1.

**Authors' reply:** Point 19 of the GOS database is shown in figure 1 with the name "GOS point". The figure 1 is self-explanatory, each vector is already associated with its meaning with source color in same color as vector. Additional information is presented in text caption.

**Referee #1:** Line 179, LiDAR DEM covers an entire area, however, the area covered in km² and in form of a map is missing in presented research.

**Authors' reply:** Agreed with Referee # 1. We added in line 76: "The study area covers the entire Piancó-Piranhas-Açú estuary, represented by a rectangle in the measurements 62 x 35.5 km, corresponding to an area of approximately 2,200 km².

**Referee #1:** Table 1, Hazard attribute values, in score 1 to 5, however score for 2 is missing. Justify why it is not considered.

**Authors' reply:** Because we only use 4 climate change scenarios in the literature (including the present), we adopt only 4 flood hazard classes. The most aggravating and least likely scenario (RCP8.5) received the highest score (score 5), as it is the one with the greatest danger.

**Referee #1:** Table 2, Vulnerability values, in score 1 to 5, however, the score for 0 is missing. Justify why it is not considered or line 216 Score from 1 instead it will start from Score 0.

**Authors' reply:** We agree with Referee # 1 and changed line 244 to mention the class "0" (in the vulnerable class).

**Referee #1:** Results and Discussion: What is the significance of the use of High-resolution DEM (LiDAR) data in this work?

**Authors' reply:** We added in line 249... The flood risk mapping from high-resolution DEM provides the knowledge to optimize investments and provide flood risk management in high accuracy (Schröter et al., 2018). The standardized altimetric reference at centimeter intervals become essential, especially when there is an interest in the analysis of land use and cover, or when economic activities occur in these environments, in order to estimate risks with accuracy (Aguiar et al., 2019).

**Referee #1:** Why the flood depth map is not presented in this study?

**Authors' reply:** The map in figure 6 is closely related to "flood depth map", since each hazard class is related to the flood tide quota for that class. In addition, in figure 8, we present an excerpt of the modeling of a tidal flood event occurred in 2015, followed by a photographic field record for model validation.

**Referee #1:** Flood waves are dynamic quantities, however, the entire concept presented in static condition. If dynamic tidal wave model or unsteady tidal flood modeling will be performed considering the river flooding and rain flooding will generate the different flood inundation or risk scenario.

**Authors' reply:** Yes. Really, if the tidal flood modeling will be performed considering this sum (river flood + tidal flood) will generate the different flood inundation or risk scenario. However, it is important to emphasize that our goal is to realize only risk of "tidal flooding". Once, that is event that has been causing major problems. In addition, it is important to mention that the study area is inserted in a sheltered coastal region dominated by tides and not by waves. (as presented by Vital et al., 2016). Therefore, we take the opportunity to add the following sentence in line 90: "The study area is inserted in sheltered coastal region, dominate by tide-modified to tide-dominated beaches (Vital et al, 2016)".

**Referee #1:** Line 269, Table 5.5, However it does not match with the table numbers.

**Authors' reply:** We accept and correct the call in the text. The correct is "Table 5".

**Referee #1:** References: Line 309, reference 1 should cite in the English language.

**Authors' reply:** We quote the title of the work according to originality of publication. As it was published in Brazilian Portuguese, we keep the original title. Natural procedure, as in other articles published by the journal *Natural Hazards and Earth System Sciences.*

**Referee #1:** Page 17, Photographs A to C, shows the tidal flooding in different years i.e. 2011, 2015, what is the relation with the scenario generated. How it will be utilized for validation, although the scenario is generated based on the 20-year return period.

**Authors' reply:** The Figure 2, present on page 19, is associated with paragraph on page 4 (study area), where we relate the figure to recent tidal flood events in study area. We believe that it is extremely important to contextualize the reader to problem at focus (tidal flood in north coastal of Rio Grande do Norte state).

**Referee #1:** These are the major deficiencies observed in this work and need to improve. I rated this paper as a major revision; Happy to review a revised version with significant improvement.

**Authors' reply:** We would like to thank you for your constructive comments and for taking the time to critically review our manuscript.

**Additional relevant changes:** We take the opportunity to update the citation on climatic aspects of study area in the paragraph that starts at line 80.

---

## Author Comment (AC3) · 11 Jul 2020

**Response to General Comment:**

**Referee #2:** General comments: I have just finished reviewing the manuscript entitles 'Tidal flood area mapping fronts the climate change scenarios: case study in a tropical estuary of Brazilian semiarid' by Araújo et al. Overall the deals with a very interest subject in an area with little published information. I believe that the article is of interest and within the scope of the journal but in my opinion needs significant improvements before publishing both methodological and scientific (see detailed comments). The most important of the problems is that the hazard methodology and classification is not presented correctly (looks confusing) and it needs further improvements. The data and methods used are presented in a way that is not easy to judge is the analysis used is correct.

**Authors' reply:** Many thanks for review; we are pleased that the reviewer finds the manuscript to be interesting and relevant. We would like to thank you for your positive remarks and for taking the time to criticize positively our manuscript.

**Referee #2:** Abstract and Introduction sections very general and they do not help the reader to understand the processes related. Finally, the objective presented at the end othe Introducion are rather technical. Study area has a lot of information that is not necessary or related with the objectives of the article an results and discussion need further work.

**Authors' reply:** We analyzed all considerations and restructured the text of manuscript to solve the questions mentioned by Referee #2. However, we believe that the goal of work is not merely technical, but also scientific. Since the mapping of tidal flood risk will be the result of a complex analysis, adopting unprecedented information, and will support an update art study of tidal flood and also bring key concerns in the study area to policy makers and decision takers priorities.

**Referee #2:** Abstract 'Previous studies on tidal flood mapping are mostly with continental and/or global scale approacher' this statement is not correct. There are far more local scale flooding studies, it is true that the impact of the global or continental wide studies tends to be higher but the number of studies is not.

**Authors' reply:** Respectfully, we disagree. Although the number of studies on "floods" (generic term) are larger at the local scale, specific studies on "tidal floods" are scarce, mainly in Brazil and other South American countries. In greater numbers, there are some more in a continental and/or global approach.

**Referee #2:** Introduction LINE 26: 'has occurred at an accelerating rate' is occurring, as it continues to occur.

**Authors' reply:** We agree with Referee # 2 and changed "During the last quarter-century, the global mean SLR has occurred at an accelerating rate, averaging about +3 mm/year" by "The SLR global mean is occurring at an accelerating rate, averaging

about +3 mm/year (recorded for during the last quarter-century)". We changed "flood risk" by "tidal flood risk" in abstract (line 3).

**Referee #2:** LINES 33-34: 'Decades ago, the flooding usually happened only during a powerful or localized storm now can happen when a steady breeze or a change in coastal current overlaps with high tide (NOAA, 2019).' This statement is only valid for the US area and for 'sunny day flooding' no storm conditions. It cannot be extrapolated to other areas since local environmental and infrastructure (drainage system) parameters are important. Please rephrase. Also explain in the introduction with what type of flooding this work is focused marine flooding, flooding through the drainage system or both.
**Authors' reply:** We agree with Referee # 2 and changed to "Decades ago, the flooding usually happened only during a powerful or localized storm now can happen when a steady breeze or a change in coastal current overlaps with high tide, as occurs for example in USA (NOAA, 2019)".

**Referee #2:** LINE 35: 'In Brazil, the current panorama of coastal flooding is extremely worrying (Losada et al., 2013).' In what terms, this statement is not supported by the paper. Please explain which are the worrying factors according to Losada et al. Since this is a local study please provide information related with the specific study area.
**Authors' reply:** We agree with Referee # 2 and changed the sentence.

**Referee #2:** LINE 55: the objective as it expressed looks more technical than scientific. Please try to reformulate it.
**Authors' reply:** We agree with Referee # 2 and changed the sentence: "the objective of this work was develop and apply a methodology for tidal flood risk mapping, given the current scenarios of rising sea level trends, adopting as a case study the Piranhas-Açú Estuary, the northern portion of the state of Rio Grande do Norte, Brazil".

**Referee #2:** Study Site: The first and the third paragraph have a lot of repeated information (in the first one without proper referencing). Please remove.
**Authors' reply:** We agree with Referee # 2 and changed the sentence.

**Referee #2:** LINES 82-87: The tidal information is not properly provided. What is the mean tidal range, the mean high water spring tide and the mean high water neap tides? The reduction level (RT) is a national level that is not explained until the methods section. Please refer ranges relative to MSL.
**Authors' reply:** The Hydrography and Navigation Directorate (DHN) of the Brazilian Navy (MB), the institution responsible for maritime monitoring, adopts the so-called reduction level (RL) as a reference for maritime quota. The RL is level that corresponds to the average of low tides of syzygy, to eliminate the variations of tides and to guarantee to navigator that it does not find any depth less than those represented in the nautical chart. We added the information in updated manuscript.

**Referee #2:** LINE 103: 'been suffering lately' what do you mean by lately? Since when? How many events per year?

**Authors' reply:** Recent reports, from the years 2010, were found in local newspapers and blogs, as well as personal observation *in loco* (personal communication). Since then, frequently, due to the effects of the hydrodynamic forcing, streets and houses are flooded; bringing harmful consequences to several communities of the northern coast of Rio Grande do Norte, and many other places in Northern Brazilian coast. We added the information in updated manuscript.

**Referee #2:** LINE 116: It is not possible to have a one line section. Please provide more information on the tidal data. Length of the record, precise location, gaps in the record frequency of data recording, reference level and data treatment.
**Authors' reply:** The sentence is actually just a presentation for the next subtopics. The goal was that the sentence was short and to point. We decided to keep the sentence.

**Referee #2:** LINE 120: provide model resolution and other models parameters. Any indicators on the ability of this model to predict correctly the observed storm surge? Please provide calibration and validation information.
**Authors' reply:** The meteorological tidal GOS database consists of a time series selection of regional reanalysis along the Brazilian coast, numerically simulated with the three-dimensional circulation model ROMS (Regional Ocean Modeling System). ROMS consists of sea level variation data due to atmospheric factors. To generate the series, the barotropic module (2DH) of the model in global mesh with 0.25° spatial resolution and bathymetry data of ETOPO2 model (NOAA) was used. As forcing the model, pressure data at sea level and global winds (10m high) from Reanalysis 1 (NCEP / NCAR) were used. The wind and pressure data have a spatial resolution of 1.9° and a 6-hour temporal resolution, covering the period from 1948 to 2008. We added the information in updated manuscript.

**Referee #2:** LINE 124: No need to define the Astronomical tide
**Authors' reply:** Despite being a widespread concept, we believe it is pertinent to maintain the sentence.

**Referee #2:** LINE 130: It is not clear if the data are measured or predicted water levels. If the data are max and min measured then they are not purely astronomical data but they have also meteorological and judging by the figure 1also river components. If on the other hand the data are predicted max. and min. values from tidal constituents then there is no reason to calculate the return period of the data there is no probabilistic part only deterministic. Please describe better the data and adjust the statistical methods used in the text.
**Authors' reply:** For this job, the Astronomical Tide data are the result of deterministic forecasts prepared and provided by the Brazilian Navy's Directorate of Hydrography and Navigation (DHN). The maximum annual astronomical tide level was used from the astronomical tide forecast data released by DHN. We added the information in updated manuscript. Despite being a deterministic method, the return period inference (probabilistic inference) is used in the final model, since it will be used the tidal behavior parameter and not the raw tide data.

**Referee #2:** LINE 141: see comment above.

**Authors' reply:** We answered above.

**Referee #2:** LINE 153: change 'payback' with 'return'
**Authors' reply:** We accepted and modified in manuscript.

**Referee #2:** LINE 195: 'Church et al., 2013)' not in the reference list. Also is this a global MSL estimation? Since it is a local study a local estimate would be better. You do not explain where you use the climate change data. Their use in Table 1 is confusing since extreme climate change scenarios are related with low hazard level. In general the use of CC scenarios in Hazard levels is confusing.
**Authors' reply:** We add reference Church et al. (2013). Regarding MSL estimation, we used the scenarios found in literature. RCPs are scenarios with a global projection, whereas the IBGE scenario is a projection at a regional level (for Brazilian northeaster). The latter result of reports on data from IBGE tidal network. IBGE is the Brazilian Institute of Geography and Statistics.
Each rate of sea level rise in scenarios used is due to robust modeling of sea level projection under face climate change.
In fact, in many studies the largest projections correspond to greatest risk and/or danger. The weights assigned to each scenario are associated with the MSL rise that projection establishes. However, for our study, the highest weights correspond to small rise, since the probability of occurring is more palpable (probability to occurring), that is, the level at which that danger already occurs. When we increase an MSL projection, we decrease the weight, since the probability of that variation occurring is more uncertain. Therefore, the greatest dangers correspond to areas with the lowest rates and the lowest dangers correspond to areas with the highest rates. It is a logical and pertinent thought for flood hazard mapping, as seen in Araújo et al (2019).
We restructured the text of manuscript to clarify the mentioned questions.

*Araújo, P. V. N., Amaro, V. E., Silva, R. M., and Lopes, A. B.: Delimitation of flood areas based on a calibrated a DEM and geoprocessing: case study on the Uruguay River, Itaqui, southern Brazil, Nat. Hazards Earth Syst. Sci., 19, 237–250, https://doi.org/10.5194/nhess-19-237-2019, 2019.*

**Referee #2:** LINE 205-210: It is no clear on how you select your hazard levels and also which is your hazard indicator (see Ferreria et al., 2017 Process-based indicators to assess storm induced coastal hazards. Earth-Science Reviews 173). It is always better to select a proxy that is a measure of hazard (e.g. inundation depth instead of total water level.).
**Authors' reply:** We explained previously. For this study, the highest weights correspond to small rise, since the probability of occurring is more conspicuous (probability to occurring), that is, the level at which that danger already occurs. When we increase an MSL projection, we decrease the weight, since the probability of that variation occurring is more uncertain. Therefore, the greatest dangers correspond to areas with the lowest rates and the lowest dangers correspond to areas with the highest rates. It is a logical and pertinent thought for flood hazard mapping.

**Referee #2:** Table 1 is not easy to understand as RCP8.5 is a low hazard scenario. In general this section is not clear and needs more work. Normally hazard can take a value of 0 'no hazard'.

**Authors' reply:** The highest weights correspond to small rise, since the probability of occurring is more palpable (probability to occurring), that is, the level at which that danger already occurs. When we increase an MSL projection, we decrease the weight, since the probability of that variation occurring is more uncertain. Therefore, the greatest dangers correspond to areas with the lowest rates and the lowest dangers correspond to areas with the highest rates.

**Referee #2:** LINE 212: The vulnerability is actually land use vulnerability. Which were the parameters used for the classification. Economic importance or other? The classification was based of regional or national stakeholders or was made by the authors?

**Authors' reply:** We added a new topic in updated manuscript: "3.5 Land use and cover map".

**Referee #2:** LINES 230-237: It is not easy to observe trend with relative short time-series. Have you checked if there is a correlation with any of the large weather patterns and indices that affect the area? It is important to explain what kind of tidal data you present see my comment in methods section. Is it possible to extend the surge and tidal data to cover the events you have measurements?

**Authors' reply:** We understand. Unfortunately, the temporal series of meteorological tide data is limited 2008´s. We believe that after this period, the meteorological tide has been influencing in the phenomenon of tide flooding more strongly. However, for this work it will not be possible to build the data. We are in the process of expanding this time series with several researchers, but it will take some time to complete (around years). Corroborating for publication this manuscript. Soon this work becomes the pioneer of many that will come on the phenomenon of tidal flooding in this study area, and we will mention this research in near future as an initial starting point of investigations.

**Referee #2:** LINES 240-247: this information is better in the study area.

**Authors' reply:** Yes. It would also fit the Study Area topic; however, we use the sentence as an engine of discussion.

**Referee #2:** LINES 251_301: it is difficult to comment on the results since I the hazard classification is not clear. Up to the results you do not mention which is the process of inundation (flooding form the see or drainage water). Please introduce this earlier in the text with the appropriate information of the type and location of the drainage system. Also how you do the mapping in GIS? Are you using an algorithm with hydraulic connectivity or it is a bathtub model?

**Authors' reply:** Tidal flooding is the focus of the work. However, due to the flat and low region, very close to sea level and inserted in an anastomosed estuary, the tidal flooding also occurs through the rain drainage system. Occurring the return of water through the system. We reinforce that the river mentioned in work is a tidal channel.

We updated the information in updated manuscript. The algorithm used was only a bathtub model (as mentioned in table 4).

---

## Author Response (AR2)

**Author Comments to Referee #2 (Major Revision)**

Major Revision on "Tidal flood area mapping fronts the climate change scenarios: case study in a tropical estuary of Brazilian semiarid" by Araújo et al.

Dear Referee #2 (Major Revision),

We do appreciate your constructive, thoughtful, careful, and helpful comments and suggestions. After careful discussions and analyses, we finished the preparation of responses to you. If there are any new
10 comments or suggestions, please let us know.

In this document, we respond to the comments received point by point.

Best Regards,
15 Paulo Victor N Araújo and coauthors

**Response to General Comment:**
=============================================
20 **Referee #2:** LINE 21: remove 'area'.
**Authors' reply:** We accepted and modified in manuscript.

**Referee #2:** Line 64: 'to develope and apply'.
**Authors' reply:** We accepted and modified in manuscript for "to develop and apply".

**Referee #2:** LINES 95-99: Is the reduction level the 'lower astronomical tide'? If yes please rename it since lower astronomical tide is an internationally used term for the datum used in Navigation maps. If it is not please explain what is the difference.
**Authors' reply:** No. The DHN of MB, the body responsible for maritime monitoring, adopts the so-
30 called reduction level (RL) as a reference for maritime quota. The RL is level that corresponds to the average of low tides of syzygy. It is a safety method, adopted by Brazilian Navy to eliminate tidal variations and guarantee the navigator that he does not find any depth less than those represented in the nautical chart.

**Referee #2:** LINE 100: what do you mean with 'mean quadrature of high tides'. Is this the mean neap tides? What about the mean spring tides that are more relevant to flooding conditions.
**Authors' reply:** OK, modified in manuscript.

**Referee #2:** LINE 120: remove '(type of focus flooding of this job).' The term job is not appropriate, work or article is better, but I believe it is not needed since now it is explained before.
**Authors' reply:** We accepted and modified in manuscript for "type of focus flooding of this work"

**Referee #2:** LINE 120: Reference the recent reports.
**Authors' reply:** As it is gray literature, with local information context, we prefer not to quote the references in  work.

**Referee #2:** LINES 215-220: IS the IBGE a climate change scenario or 2.1mm/year is the measured sea level rise in the area using tidal gauges? If it's the latter the two values are not representing the same conditions. The 2.1 mm/year is the current condition not a climate change scenario.
**Authors' reply:** The IBGE is a climate change scenario.

**Referee #2:** LINES 241-264: I have various consideration regarding the hazard Classes. As I stated in previous review the classes are in a reverse order. The authors reply saying that 'for this study, the highest weights correspond to small rise'; however, in LINE 247 they state 'Hazard map is the likelihood of the process occurring with magnitude M (destructive potential) and…'. First of all Hazard mapping is a spatial representation of the hazard not a value. There is a direct inverse relationship between hazard magnitude and hazard probability. If the authors want to work with hazard probability they should estimate it, but this is not hazard is hazard probability. This probability is related with a return period but the authors in Table 1 and 2 refer to Hazard and not Hazard probability. Also in Table 4 levels were provided but they were not converted to probability. A key parameter for that is determining a threshold of water level in which damage is considered.
Also the combinations of SLR for 2100 for different RCPs for local SLR rate and return periods are confusing because different temporal scales are mixed. A hazard map should provide levels of probability for a fixed time period.
**Authors' reply:** In fact, it is a probability of hazard. We call it hazard, but we make it clear ... "Hazard map is the likelihood of the process occurring with magnitude M (destructive potential) and…".

**Referee #2:** LINE 64 the authors mention that develop and apply a method; however no method development is presented in the paper just an application. If a new methodology is developed the stages and how the weight are obtained needs to be explained.
**Authors' reply:** We accepted and modified in manuscript.

**Referee #2:** LINES 275-294: Regarding the return period of the tidal level, I believe that the application of an extreme value statistic in a deterministic parameter like the tidal level is an error. The authors state that they use the tidal behavior parameter. I am not sure what this means; the authors will need to explain this in the text. I am just stating that even in Figure 5B it is obvious that the tidal level has a maximum at 2.85, the tidal levels will repeat themselves every 19.8 years so for a return period of 20 years you just need to select the maximum tide recorded the past 19.8 years.

**Authors' reply:** Despite being a deterministic method, the return period inference (probabilistic inference) is used in the final model, since it will be used the tidal behavior parameter and not the raw tide data.

[revised manuscript text omitted]

---

## Author Response (AR3)

**Author Comments to Referee #2 (Major Revision)**

**Major Revision on "Tidal flood area mapping fronts the climate change scenarios: case study in a tropical estuary of Brazilian semiarid" by Araújo et al.**

Dear Referee #2 (Major Revision),

We do appreciate your constructive, thoughtful, careful, and helpful comments and suggestions. After careful discussions and analyses, we finished the preparation of responses to you. If there are any new comments or suggestions, please let us know.

In this document, we respond to the comments received point by point.

We hope to have finished this stage, and we are hopeful with the acceptance of this publication.

Best Regards,
Paulo Victor N Araújo and coauthors

**Response to Comments:**
=================================================
**Referee #2**:(1) To my question if the reduction level is the lowest astronomical (LAT) tide used as common reference datum for navigation maps the authors answered No, but they explanation they provide ´RL is level that corresponds to the 30 average of low tides of syzygy´, suggest that the RL is a chart datum estimation similar to LAR or to the mean lower low water (MLLW) used by the US Navy. The importance here is to provide to the reader an idea of the reduction level is.
**Authors' reply:** OK, perfect! We accepted and add in the manuscript, in line 87 (page 3)… "The RL is a chart datum estimation similar to LAR or to the mean lower low water (MLLW) used by the US Navy".

**Referee #2**: (2) To my question if the IBGE is a climate change scenario they replied Yes but the text provided in the document ´The latter, result of reports on data from IBGE tidal network ´ suggest that IBGE projection is not a climate change scenario but the measured sea level change in the region. Both can be used but the IBGE data is a completely different estimation for sea level rise for the year 2100. It is not a better estimator it is a simple linear projection of the past measurement.
**Authors' reply:** Correct, we agree! The IBGE is a simple linear projection based on data variation obtained by tidal gauge. Modified in manuscript, in line 207 (page 7) for "The IBGE scenario is result from a simple linear projection based on data variation obtained by tidal gauge, while IPCC scenarios are results from modelling robust of sea level projection under face climate change".

**Referee #2**: (3) The line 207: ´In all rate of sea level rise in scenarios used, are from robust modelling of sea level projection under face climate change ´ is grammatically incorrect and should be changed.

**Authors' reply:** We accepted and modified in manuscript for "The IBGE scenario is result from a simple linear projection based on data variation obtained by tidal gauge, while IPCC scenarios are results from modelling robust of sea level projection under face climate change".

**Referee #2**: (4) As for the use of a return period statistics to the Astronomical tide I am still not convinced. The only uncertainty about the astronomical tide in the future is not the maximum level but the timing of the astronomical tide related with the meteorological tide. Also, as it can be observed in Figure 5B the fitting applied (linear) is not a good explanation for the data that, as expected, present a level of the maximum tidal levels after the 5 years return period value. A combined extreme value analysis of the total level (astronomical+meteorological) has a purpose.

**Authors' reply:** In fact, the referee's logical reasoning is well-founded and extremely acceptable. However, the principle of working with the tidal return period is to show that the tides with a 20-year return period (which are much smaller than the maximum heights and are more likely to occur) result in a representative flood hazard zone in study area. That is, to highlight the flood hazard that are commonly occurring in the region. So, we decided to stick with the initial strategy.

=================================================

---

## Author Response (AR4)

**Author Comments to Editor (Minor Revisions)**

**Minor Revisions on "Tidal flood area mapping fronts the climate change scenarios: case study in a tropical estuary of Brazilian semiarid" by Araújo et al. (nhess-2020-92).**

Dear Editor (Minor Revisions),

We do appreciate your constructive, thoughtful, careful, and helpful comments and suggestions. After careful discussions and analyses, we finished the preparation of responses to you. If there are any new comments or suggestions, please let us know.

In this document, we respond to the comments received point by point.

We hope to have finished this stage, and we are hopeful with the acceptance of this publication.

Best Regards,
Paulo Victor N Araújo and coauthors

**Response to Referee #2 Comments:**
==============================================
**Referee #2**: (1) An English native speaking revision is mandatory.
**Authors' reply:** We accepted and performed a substantial revision of the English language throughout the text of the manuscript with the support of an English-speaking professional.

**Referee #2**: (2) line 60: the introduction is well written and argued; I suggest the author consider to add a sentence to describe better the topic of the manuscript (e.g. the dataset used and what elements of this paper are innovative and should be evaluated by readers).
**Authors' reply:** We accepted and add in manuscript, in line 64: "This work has an innovative character, combining a robust data set in an integrated spatial analysis, and which must be evaluated by readers".

**Referee #2**: (3) Line 68: in this manuscript, I found several problems related to incorrect use of the language; for this reason, I strongly recommend a revision by English native speaking. For example, I am not sure about the meaning of "rainy late to autumn". I do not think that the reviewer's duty is the revision of the language of the manuscript, but I believe that a substantial revision is mandatory in this paper.
**Authors' reply:** We accepted and performed a substantial revision of the English language throughout the text of the manuscript with the support of an English-speaking professional. The cited expression has been corrected to (line 70): "rainy season in late autumn and early winter".

**Referee #2**: (3) Line 110: in figure two, are mentioned several municipalities. I suggest the author add some information about the areas involved in tidal floods that are mentioned in this part of the manuscript. I suppose, for example, that the flooded areas are part of Porto de Mangue and Macau municipalities.

**Authors' reply:** We accepted and add in the text legend of figure (Figure 1): "The black dotted line represents the geopolitical boundaries of the municipalities that make up the study area".

**Referee #2**: (4) Chapter 3.4: the description of adopted scenarios seems to be too synthetic; I suggest the authors revise the chapter and describe better the adopted dataset.

**Authors' reply:** We understand the suggestion, however we understand that the topic, despite being synthetic, is well understandable.

**Referee #2**: (5) Figure 6: it is not easy for readers to have an idea of the position of the city of Macau mentioned in the description of figure 6b and 7. Figure 6b: in the lower part of the image, there is the river (in black), but the river disappeared in the corner of the image that is red. I do not know if this is an error or some infrastructure covers the river.

**Authors' reply:** We don't agree. The vectorization of the river only extends to the meeting of the first bridge. Therefore, after the bridge, the color red is indicated.

**Referee #2**: (6) Figure caption: I suppose, as mentioned in the text, that the cemetery area cannot be flooded, according to the model; I think that this important result should be mentioned in the figure caption because this area does not belong to an extremely low flood hazard level but another (not mentioned) category.

**Authors' reply:** We don't agree. Precisely because the flood hazard area is not included, we understand that the area cannot be classified.

**Referee #2**: (6) Figure 7: I not sure that this image is important and representative.

**Authors' reply:** We understand that the image represented in Figure 7 represents a very important gain in understanding the problem.

**Referee #2**: (7) Line 295: "It is important to mention that the land on which the local cemetery is in the urban area of Macau is one of the few urban sectors in the city not to suffer from tidal flood scenarios." This result is interesting, but I think that the description is too limited. I believe that readers want to know if the area is not affected by the presence of a local topographic effect or other elements.

**Authors' reply:** We accepted and add in manuscript, in line 299: "This result becomes extremely important for the future urban planning of the city of Macau".

**Referee #2**: (8) Figure 9: I do not understand why this map is presented at the end of the paper. The map of land use is presented at the end of the paper, but it is mentioned for the first time in chapter 3.

**Authors' reply:** We took the logical sequence to the letter: understanding the hazard, understanding the vulnerability (result of the land use and cover map) and finally, understanding the risk.

**Referee #2**: (9) Lines 295 – 305: in ten lines, the authors mentioned three figures (9-10-11) without a real explanation of the obtained results. I do not think that figure 9 is really necessary at the end of this chapter ad that, if the authors want to present it, they could do it before.

**Authors' reply:** We don't agree. We believe that figure 9 is extremely important and is placed under a logical sequence. We took the logical sequence to the letter: understanding the hazard, understanding the vulnerability (result of the land use and cover map) and finally, understanding the risk.

**Referee #2**: (10) An accurate description of the results proposed in figure 10 is missing. The sentence of line 304 cannot be considered acceptable.

**Authors' reply:** We accepted and add in manuscript, in line 304: "…ranking the of…".

**Referee #2**: (11) Chapter 4.3: this is the most important result of the paper, and the author described it in 5 lines!

**Authors' reply:** We accepted and performed a substantial add in the subtopic.

**Referee #2**: (12) As suggested in figure 10, this part of the manuscript should be explained better.

**Authors' reply:** We accepted and performed a substantial add in the subtopic.

===============================================

---

## Author Response (AR5)

**Author Comments to Editor (Minor Revisions)**

Dear Editor (Minor Revisions),

We do appreciate your constructive, thoughtful, careful, and helpful comments and suggestions. After careful discussions and analyses, we finished the preparation of responses to you. If there are any new comments or suggestions, please let us know.

We have attached the manuscript with the tracking of the changes, after a robust English language review with the support of an English-speaking professional.

Regarding the figures, all were updated to the 600dpi version, and the quality was observed in the digital version manuscript (not printed).

About the cemetery region, as it is outside the risk classes, we understand that it is automatically in an area without risk of flooding.

We hope to have finished this stage, and we are hopeful with the acceptance of this publication.

Best Regards,
Paulo Victor N Araújo and coauthors